# A cytotoxic T cell inspired oncolytic nanosystem promotes lytic cell death by lipid peroxidation and elicits antitumor immune responses

Zhigui Zuo[1,4], Hao Yin[2,4], Yu Zhang[3], Congying Xie[3] & Qinyang Wang [2,3] ✉

Lytic cell death triggers an antitumour immune response. However, cancer cells evade lytic cell death by several mechanisms. Moreover, a prolonged and uncontrolled immune response conversely leads to T-cell exhaustion. Therefore, an oncolytic system capable of eliciting an immune response by killing cancer cells in a controlled manner is needed. Here, we establish a micro-scale cytotoxic T-cell-inspired oncolytic system (TIOs) to precisely lyse cancer cells by NIR-light-controlled lipid peroxidation. Our TIOs present antigen-based cell recognition, tumour-targeting and catalytic cell-lysis ability; thus, the TIOs induce oncolysis in vivo. We apply TIOs to preclinical cancer models, showing anti-tumor activity with negligible side-effects. Tumour regression is correlated with a T-cell based anti-tumour immune response and TIOs also improve responses to anti-PD-1 therapy or STING activation. Our study provides insights to design oncolytic systems for antitumour immunity. Moreover, activation of STING can reverse T-cell exhaustion in oncolysis.

Cell death is a vital reason for triggering inflammation and immune responses[1,2]. Lytic cell death, featuring plasma membrane rupture, can release numerous proinflammatory molecules to reshape the microenvironment and activate the immunity[3–5]. For malignant cells that exhibit sufficient antigenicity, the inflammatory signals resulting from cell lysis may trigger antitumour immune function[1,2,6–8]. Recently, we and others identified pyroptosis, a classical lytic cell death, can evoke antitumour immunity[6–8]; thus, lytic cell death-mediated antitumour immunotherapy is valuable for both biomedical and clinical investigations. However, tumours have evolved various mechanisms to avoid lytic cell death and escape the antitumour immune activation[3,9–11]. In nature, cancerous cells always negatively regulate lytic cell death by downregulating the expression of executive proteins (e.g., gasdermin in pyroptosis)[9] or upregulating expression of proteins involved in membrane repair functions (e.g., ESCRT-dependent membrane repair, Fig. 1a)[10,11]. Furthermore, any lytic cell death is controlled by complex and programmed signalling networks, but the potential crosstalk involved in these networks may lead to disruption of cell death or conversion to immunologically silent variants (e.g., apoptosis)[1–3]. Notably, even if cell lysis is successfully triggered, it is still unknown whether the corresponding immune response can exert anticipated antitumour function[3,9,12–15]. Growing evidence showed persistent uncontrolled inflammatory signals could cause cytokine release syndrome (CRS) due to hyperactive immune responses or led to immune dysfunction due to T-cell exhaustion[12–15]. These issues reflected the challenges encountered when inducing antitumour immunotherapy through oncolysis. During the past decades, a better understanding of biotechnology, chemical biology and tumour immunology has prompted the emergence of artificial oncolytic systems for mediating lytic cell death[16,17]. One of the most representative is the oncolytic virus (OV), which can selectively replicate in host cells and generate a direct lytic effect[16]. Recently, a variety of genetically modified OVs have been

[1]Department of Colorectal Surgery, The First Affiliated Hospital of Wenzhou Medical University, Wenzhou, Zhejiang, P.R. China. [2]Institute for Advanced Research, Wenzhou Medical University, Wenzhou, Zhejiang, P.R. China. [3]The Second Affiliated Hospital and Yuying Children's Hospital of Wenzhou Medical University, Wenzhou, Zhejiang, P.R. China. [4]These authors contributed equally: Zhigui Zuo, Hao Yin. ✉e-mail: wangqy@wmu.edu.cn

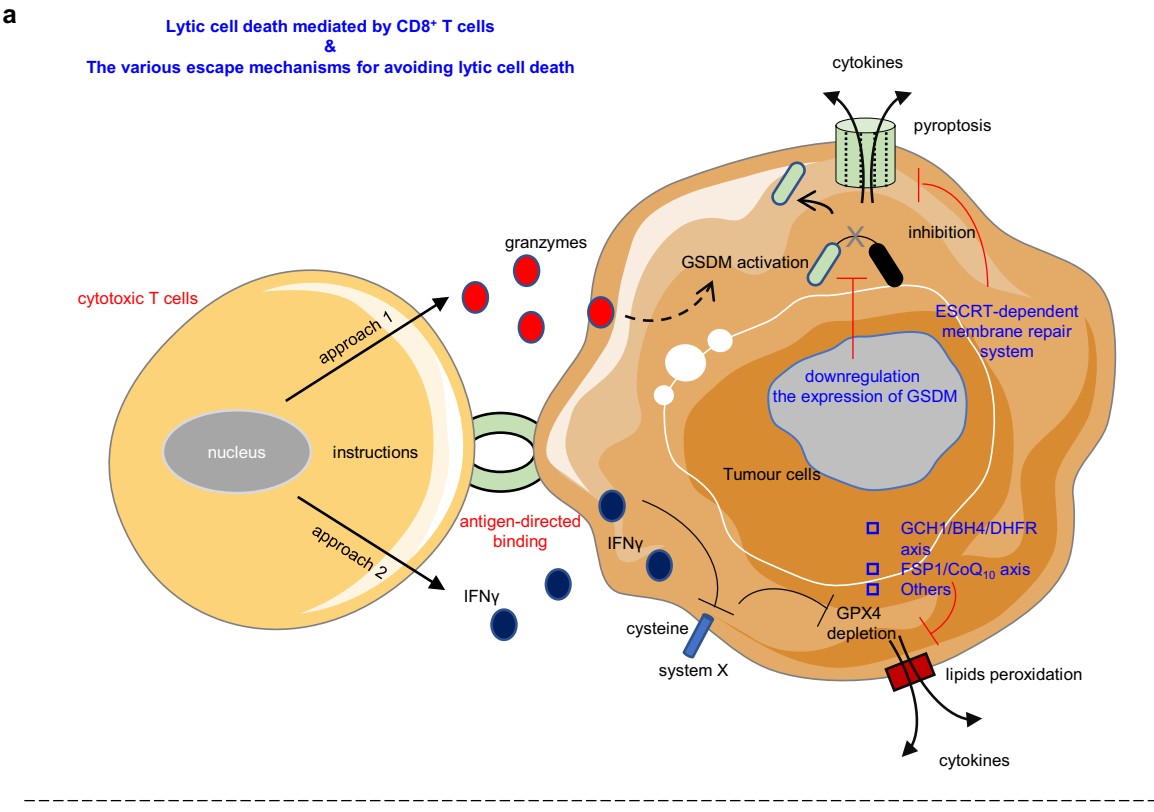

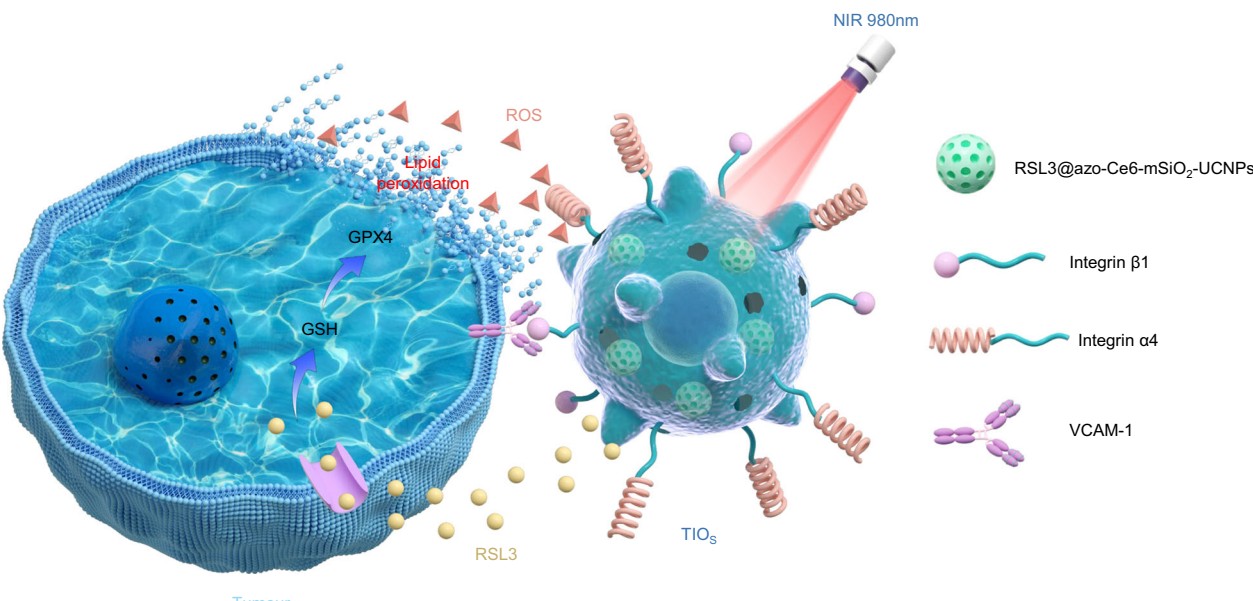

**Fig. 1 | The oncolysis mediated by CD8+ T cells or TIOs. a** The CD8[+] T cells mediated cell lysis. During the killing process, the CD8[+] T cells bound the tumour cells by antigens coordination and the signal pathways guided nucleus to prepare the cytotoxins (e.g., granzymes or IFNγ). Then, the killing tasks started and the CD8 + T cells could kill all the dangerous cells in a catalytic manner. In approach 1, the CD8[+] T cells released granzymes would enter the tumour cells, trigger the cleavage of GSDMs and cause the pyroptosis by gasdermin pores. However, the human tumour cells always downregulated the expression of GSDMs and the membrane-repairs mechanisms were ready to quench the pyroptosis. Moreover, in approach 2, the CD8[+] T cells released IFNγ blocked the tumour cells to take up cysteine, which led to the depletion of GPX4, lipid peroxidation and the cell lysis. However, the cellular intrinsic reductive systems (e.g., GCH1/BH4/DHFR axis) were ready to quench the oxidative stress. **b** T-cell-inspired oncolytic system (TIOs) mediated cell lysis. The TIOs bound the tumour cells by antigens coordination. Then, the NIR light would trigger the TIOs to generate ROS and release RSL3 to destroy the tumour cells plasma membrane by lipid peroxidation. After the tumour cells lysis, the TIOs would regenerate and bind other tumour cells. Thus, the catalytic killing would happen when NIR light was inputted.

developed as oncolytic agents for the induction of systemic anti-tumour immunity in preclinical or clinical research[18]. However, the lack of delivery systems, poor controllability, unsatisfactory tumour specificity, and instable clinical effect reduced the reliability and safety of the above oncolytic tools[18]. As a great addition to these OVs, the sophisticated amphiphilic membranolytic small molecule (MLSM) could destroy the cellular plasma membrane independent of any biological mechanisms[19]. However, the MLSM system not only did not solve the problems of OVs, but also required a higher dose of consumption to mediate cell lysis[19]. To date, it is still lacking a method capable of breaking through natural restrictions to effectively mediate lytic cell death.

The polyunsaturated fatty acid-containing phospholipid (PUFA-PL) is a main component for maintaining the stability of cell plasma membrane, the extensive oxidization of PUFA-PLs leads to disintegration of the cytoskeleton and cell lysis[20–22]. Recently, cytotoxic T cells (CD8[+] T cells) were revealed to indirectly mediate PUFA-PLs peroxidation and oncolysis (Fig. 1a)[23]. As known, the CD8[+] T cells attacked the dangerous cells in a catalytic and intelligent manners (Fig. 1a)[24]. In the battles between CD8[+] T cells and cancer cells, a catalytic amount of CD8[+] T cells actively localized to the tumour cells and performed killing tasks continually until all dangerous cells were eliminated[24]. However, the oncolysis by natural CD8[+] T cells or the engineered CAR-T cells was subject to many natural mechanism limitations, such as immune checkpoints[25]. Moreover, this oncolysis was uncontrolled and thus easily led to CRS[12,26,27]. Most importantly, the lipid peroxidation and oncolysis mediated by CD8[+] T cells was indirect, which was negatively affected by many factors and thus low-efficient[23]. Therefore, to better manipulate oncolysis-induced antitumour immune function, a T-cell like tumour-targeting oncolytic system capable of breaking through natural limitations to specifically dissolve cancer cells in a catalytic and controllable manner by lipid peroxidation is needed.

Here, we show a readily prepared near infrared (NIR) light controllable micron-scale T cell-inspired oncolytic system (TIOs) with catalytic, controllable and direct membranolytic capacity to manipulate oncolysis, inflammation and antitumour immune function (Fig. 1b). The TIO system is derived from a chemically engineered macrophage system with a liquid nitrogen freezing modification[28]. Our chemical engineering approach is based on an easily prepared Ce6-labelled upconverting nanoparticle (UCNP)-engineered macrophage system. Meanwhile, the addition of rapid liquid nitrogen freezing is used to turn off the stress responses of the nano-engineered living macrophages and reserve the antigens for targeting. As TIOs are 10-micron system, they are not engulfed in the target cells and perform the killing tasks outside the target cells, which is very similar with CD8[+] T cells and different with nanomedicines. We find that the TIO system can bind tumour cells through antigen-based recognition, followed by photocatalysed plasma membrane lysis. Further studies show that the lysis mechanism and cytokine secretion are dependent on reactive oxygen species (ROS)-mediated PUFA-PL peroxidation. Compared to other oncolytic systems, the light-driven membranolytic TIO system exhibits superior plasma membrane specificity and highly controlled catalytic killing capacity, positioning the TIO system as a reliable tool for tuning the efficiency of lytic cell death or the intensity of cytokine release in vitro and in vivo. In tumour-bearing mice, TIOs show tumour selectivity, good penetration and specific oncolysis. We are applying the TIO system to oncolysis-induced antitumour immunotherapy, which shows that tumour grafts, especially for tumours with lytic cell death escape mechanisms, are cleared efficiently with negligible damage to major organs. Tumour regression is correlated with the interaction between oncolysis-mediated inflammation and T-cell-based antitumour immune responses. Due to the tunability and specificity of lytic cell death mediated by the TIO system, we further identify that the generation of potent antitumour immunity and immune memory requires sufficient oncolysis to induce acute inflammation, as low-intensity lysis leads to T-cell exhaustion and tumour progression. Synergism with either checkpoint blockade or cGAS-STING activation can rescue immune dysfunction, but STING activation unleashes a more potent immune boost by reengineering tumour-infiltrating T cells to induce durable antitumour immunity. Our study provides insights into the design of oncolytic systems for oncolysis-mediated antitumour immunity. Furthermore, our application suggests that the tuning of reciprocal regulatory loops between inflammation and immunity plays a critical role in the oncolysis-induced immune response, and that cGAS-STING activation holds promise for maintaining antitumour immunity.

## Results

### Design, generation and properties of the TIO system

Inspired by structure and killing mechanism of the cytotoxic T cells (Fig. 1a)[29], a typical NIR light-controlled TIO could be divided into two parts, one was the functional granules to work as the killing-machine to mediate the photocatalyzed peroxidation reaction, the other was a carrier capable of storing these granules and recognising the cancer cells (Fig. 1b). Here, we designed the liquid nitrogen-posttreated mesoporous silicon-coated UCNP-engineered macrophages to work as TIO system. The general procedure was described below: the photosensitizers Chlorin e6 (Ce6) and azobenzene were decorated into the mesopores of the synthesized silicon-coated upconverting nanoparticles (UCNPs); then, RSL3, an inhibitor of glutathione peroxidase 4 (GPX4) which can reduce the cellular intrinsic reductivity to promote the peroxidation reaction[22], were loaded into the mesoporous by electrostatic adsorption. The fully functional particles were termed as RSL3@azo-Ce6-mSiO₂-UCNPs and worked as the above-mentioned granules. The production of RSL3@azo-Ce6-mSiO₂-UCNPs was subject to strict quality control, including control of particle size, zeta potential, adsorption capacity, loading capacity and light responsiveness (Supplementary Fig. 1a–h). In RSL3@azo-Ce6-mSiO₂-UCNPs, the UCNPs can generate ultraviolet (UV) light by upconverting 980 nm NIR light (Supplementary Fig. 1d), while UV light triggers Ce6 to yield ROS and azobenzene to vibrate (Supplementary Fig. 1e–h). The vibration of azobenzene promotes the release of ROS and RSL3 (Supplementary Fig. 1k), the released ROS and RSL3 would mediate lipid peroxidation reaction. After obtaining the high-performing RSL3@azo-Ce6-mSiO₂-UCNPs, we started to construct the precursors of TIOs. In this step, macrophages were selected to work as carriers for loading these granules, owing to they not only have the phagocytic function, but also have the ability to recognize cancer cells[30–33]. A series of screenings showed that the coincubation of RSL3@azo-Ce6-mSiO₂-UCNPs and macrophages (Raw 264.7 or BMDMs) for 4 h could generate ideal UCNP-engineered macrophages by balancing the loading capacity and cell viability (Supplementary Fig. 1i). Finally, to quench the side effects caused by the metabolism and stress of nano-engineered living macrophages, the UCNP-engineered macrophages were treated with liquid nitrogen for 24 h and then washed twice with PBS to obtain the TIOs.

The TIO system is porous microsphere (average size: 10 μm), presenting high stability, dispersion and suspension ability (Fig. 2a–f and Supplementary Fig. 1l). TEM image clearly showed RSL3@azo-Ce6-mSiO₂-UCNPs were clustered in the interior of TIOs, moreover, the nuclei of TIOs were still visible, and damages in the cell membrane could be seen (Fig. 2b). Different with nano-engineered living cell system, the TIOs have unique advantages. In terms of safety, although the interactions between nanoparticles and living macrophages would cause the secretion of inflammatory cytokines for a long time (Supplementary Fig. 1j), liquid nitrogen treatment resulted in the formation of ice crystals inside or outside of living macrophages, which leads to the pores in the TIOs (Fig. 2a, b). Thus, the TIOs could be washed and purified to a level where little cytokines can be detected (Supplementary Fig. 1j). After intravenous injection of TIOs into mice, the

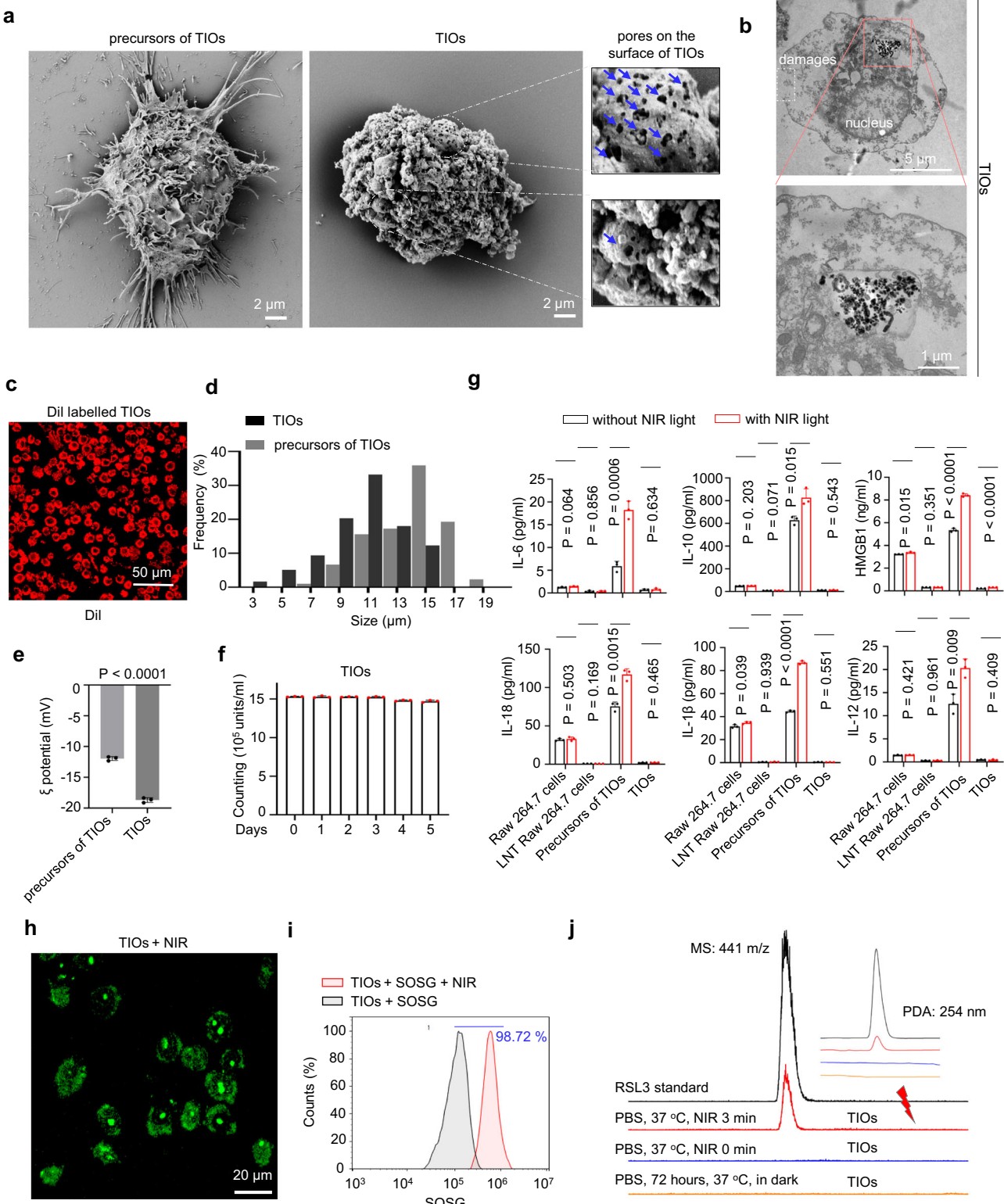

results showed the release of IL-6, white blood cells, red blood cells, platelets, body weight and body temperature of mice remained normal, indicating TIOs themselves were safe and non-proinflammatory (Supplementary Fig. 2e–k). Moreover, as liquid nitrogen post-treatment has shut down all the metabolic pathways in carrier macrophages, thus, although the TIOs was designed as a "ROS-producing factory", no ROS-related stress responses were found in TIOs (Fig. 2g). In terms of responsiveness, both the NIR light-triggered ROS-

producing and RSL3 release are key factors for TIOs-mediated lytic cell death. The suspended TIOs were treated with or without NIR light, then the singlet oxygen sensor green (SOSG) was added, respectively. The results showed the NIR light treatment led to obvious green fluorescence in images and significant positive offsets in FACS analysis (Fig. 2h, i and Supplementary Fig. 1h). At the same time, we centrifuged the above samples, then the supernatants were collected and analysed by HPLC-MS. No RSL3 could be detected when TIOs were

**Fig. 2 | Characterization of TIOs. a** Representative SEM images of TIOs and its precursor UCNPs engineered Raw 264.7 cells (scale bar = 2 μm). The blue arrows indicate the pores on the surface of TIO. **b** TEM images of TIOs. Scale bars, 5 μm, 1 μm. **c** The DiI labelled TIOs were imaged in phosphate buffer saline at 37 °C. **d, e** The average size and zeta-potential of TIOs and its precursor UCNPs engineered Raw 264.7 cells. **d** The average particle size. **e** The zeta-potential. Data are shown as mean ± s.d. ($n = 3$ independent experiments, two-tailed unpaired Student's $t$-test, $P < 0.0001$). **f** The stability of TIOs in phosphate-buffered saline (37 °C). The quantity variation of TIOs was determined by cell counter assay ($10^5$ units / tube). Data are shown as mean ± s.d. ($n = 3$ independent experiments). **g** The stress of Raw 264.7 cells, LNT Raw 264.7 cells (liquid nitrogen treated Raw 264.7 cells), the precursors of TIOs and TIOs were determined by cytokine release assay, with or without NIR light (980 nm, 1.0 W cm$^{-2}$). Data shown as mean ± s.d. ($n = 3$ independent experiments, two-tailed unpaired Student's $t$-test). In the living cell

system (precursors of TIOs), RSL3@azo-Ce6-mSiO$_2$-UCNPs themselves interacted with living macrophages, resulting in the release of many inflammatory factors (black columns). In our study, RSL3@azo-Ce6-mSiO$_2$-UCNPs were designed as NIR light controlled nanoparticles, which produced ROS under NIR irradiation (Fig. 2h, i and Supplementary Fig. 1h record the relationship between ROS generation and irradiation time). In the case of with NIR (red columns), precursors of TIOs produced more inflammatory factors than other groups, indicating an obvious ROS-related stress responses in living macrophages. **h, i** The NIR light controlled ROS generation of TIOs. The TIOs were pre-treated with SOSG. **h** The image of TIOs after NIR light treatment. Scale bar, 20 μm. **i** Flow cytometry analysis of ROS generation from TIOs by NIR light (980 nm, 1.0 W cm$^{-2}$, 5 min). **j** A HPLC-MS assay for RSL3 release from TIOs. For **a–d, h–j**, experiment was repeated three times independently with similar results. Source data (**d–g**) are provided as a Source Data file. Source data (**j**) are provided in Supplementary Information.

suspended in dark for 72 h, while obvious RSL3 release appeared when they were exposed to NIR light (Fig. 2j), which indicted the release of RSL3 was NIR light specific and TIOs themselves have high stability in the absence of NIR light.

## The antigen-directed recognition between TIOs and tumour cells

As mentioned, the principle of TIOs-mediated cancer cell lysis is to simulate the killing mechanism of cytotoxic T cells, therefore, antigen-directed cell recognition is the basic premise for TIOs-mediated cell lysis. Therefore, after verifying the TIOs was a safe and fully functional system, we began to investigate the antigen-directed recognition between TIOs and cancer cells. In TIOs, the carriers were responsible for the recognition. The mass spectrum analysis displayed many antigens of carrier macrophages were cryopreserved after the liquid nitrogen post-treatment, especially for integrins which could tether cancer cells via VCAM-1 (Supplementary Fig. 2a)[31,32]. Moreover, the FACS analysis showed the TIOs could be labelled by the monoclonal antibody of integrins (α4 and β1), indicating the recognition ability of integrins were alive (Supplementary Fig. 2b, c). The VCAM-1 could tether macrophages to cancer cells via counter-receptor α4-integrins has been widely studied[32]. Therefore, we employed catalytic amount of or overdose suspended TIOs to co-incubate with VCAM-1 positive tumour cells, then washed twice with PBS and imaged, the results showed the TIOs could tightly bind tumour cells in both situations (Fig. 3b and Supplementary Fig. 3a). However, for some VCAM-1 negative cell lines (e.g., BDMCs, TC-1, AML12, TCMK-1), the TIOs failed to bind them (Supplementary Fig. 3b). Besides, we found the CD47-SIRPα/SHPS1 and CD24-Siglec-G still alive in TIOs, which endowed the TIOs to evade immune clearance (Supplementary Fig. 2d)[33,34].

To CD8$^+$ T cells, when the recognition happened, the cytotoxins synthesis and transmission mediated by signal pathways were exquisite and efficient[29]. Basing on the antigens directed recognition between TIOs and tumour cells, we tested the light controlled RSL3 transportation in TIOs bound 4T1 cells. As RSL3 have no fluorescence, we modified the RSL3 by displacing its chlorine atom at carbonyl α site to a fluorescein (rhodamine) and got RSL3-RhB (Supplementary Fig. 3c, $Ex/Em = 488/525$ nm). Then, the TIOs bound 4T1 cells were imaged, owing to the wide absorption band of azo-Ce6-mSiO$_2$-UCNPs (from 400 nm to 520 nm, Supplementary Fig. 1g), the TIOs showed no fluorescence before the release of RSL3-RhB. After NIR light treatment, strong red fluorescence appeared in TIOs and gradually transfected to the 4T1 cells (Fig. 3c). Then, we analysed the transportation by FACS. In TIOs bound 4T1 cells, NIR light irradiation could significantly increase the intensity of RhB-RSL3, indicating a successfully cargoes transportation has been mediated by TIOs (Supplementary Fig. 3c).

## The TIOs induced PUFA-PL peroxidation

From a chemistry perspective, the peroxidation of PUFA-PLs by ROS could be easily realized in the flask. However, in the living cells, the

peroxidation of PUFA-PLs became difficult as the intrinsic reductive system (e.g., GPX4) was responsible for reversing this process (Fig. 1a)[22]. The above NIR light controlled TIO system, which can bind the tumour cells to produce ROS and inhibit GPX4 by RSL3, may provide an effective approach for PUFA-PL peroxidation in a controllable manner (Fig. 3a).

To assess the above idea, C11-BODIPY581/591 was employed to label the 4T1 cells. C11-BODIPY581/591 is a fluorescent ratio-probe of lipid oxidation, which can easily enter membranes and exhibits good spectral separation of the nonoxidized (595 nm) and oxidized (520 nm) forms (Fig. 3e). Thus, we prepared TIOs bound 4T1 cells (labelled by C11-BODIPY581/591) and exposed them to NIR light. After irradiation, the resulting damaged 4T1 cells were imaged, as shown, the TIOs-mediated lipid peroxidation was NIR light specific (Fig. 3d). Moreover, the scope of lipid oxidation was positive correlated with the irradiation time, with increasing irradiation time, the oxidized area was gradually expanded from the surface to the interior of the cells, which suggested that the intensity of TIOs-mediated lipid peroxidation was light-controllable, and the oxidative scope could be restricted to the cell membrane by reducing the irradiation time (Fig. 3d). To explore whether NIR light itself could cause lipid peroxidation, we treated tumour cells with NIR light alone, and the results showed that NIR light alone did not cause any lipid peroxidation (Supplementary Fig. 4a). Then, the above 4T1 cells were analysed by FACS. The results showed a significant decrease in non-oxidized C11 (PE channel) and an increase in oxidize C11 (FITC channel) during NIR light irradiation, indicating that PUFA-PLs were converted to the corresponding peroxidation states (Fig. 3f). Further, the index MDA (a widely accepted marker for oxidative stress) showed an increase in NIR light-treated TIOs bound tumour cells, but the increase was abolished by the antioxidants (Fig. 3g). Finally, after treating with NIR light, the PUFA-PLs in 4T1 cells membrane were separated and analysed by mass spectrometry, the content of PUFA-PLs with different unsaturated degrees decreased significantly (Fig. 3h, i). Taking these data into consideration, the TIOs was confirmed to be a reliable tool for mediating PUFA-PLs peroxidation in the living cells.

## The TIOs can induce lytic cell death in a controllable and catalytic manner

Although the NIR light controlled TIOs could induce satisfying lipid peroxidation in targeted cell membranes, it was still needed to prove whether the oxidative damage was strong enough to cause lytic cell death. Here, EMT6 cells, CT26 cells and 4T1 cells were used to test the TIOs-mediated cell death. Notably, in these cell lines, 4T1 cells were PD-L1 positive and impossible to pyroptosis as they were GSDME and GSDMD negative (Supplementary Fig. 4b, c), thus the TIOs mediated 4T1 cell lysis need to break through multiple natural limitations. As known, the lipid peroxidation led to membrane disintegration and secrete the cellular contents, which caused cell swelling and cell bubbling[6–8,22]. After NIR light irradiation, all the TIOs bound tumour

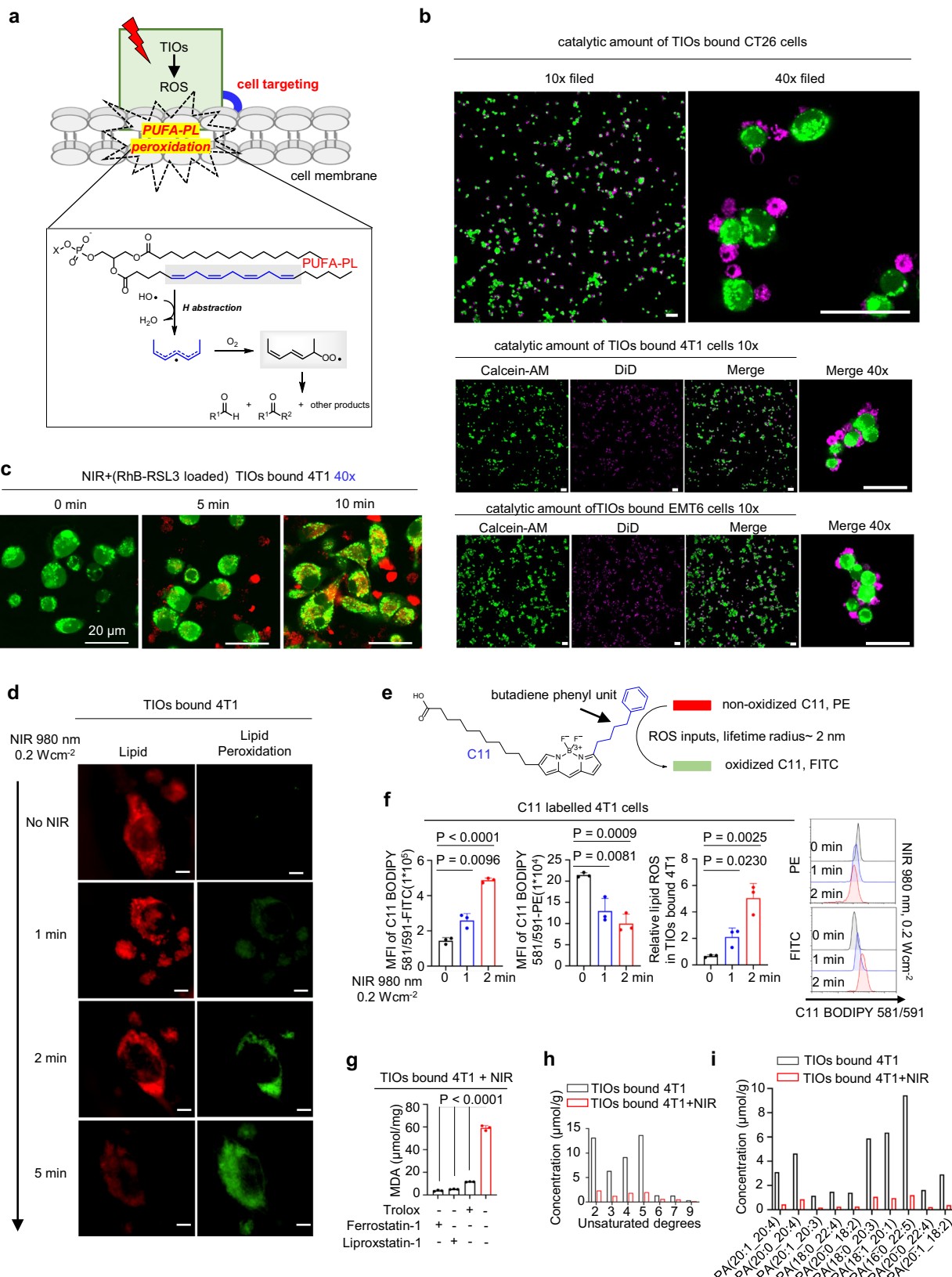

cells showed a significant morphology of lytic cell death (Fig. 4a, e.g., bubbling). Moreover, the CCK8 assay and LDH assay (a key indicator of lytic cell death) were employed to test the TIOs-mediated lytic cell death[6–8,22] (Fig. 4b). After NIR light irradiation, the results showed that lytic cell death happened in all kinds of tumour cells was positively correlated with irradiation intensity or the increasing ratio of TIOs to

tumour cells (Fig. 4b and Supplementary Fig. 4d–f), notably, both NIR light alone and the TIOs themselves were noncytotoxic (Supplementary Fig. 4g, h), moreover, as the intensity of NIR light was only 1.0 $Wcm^{-2}$, no significant thermal effects were found (Supplementary Fig. 4i). A time-lapse video showed when the TIOs bound tumour cells were exposed to NIR light, the lytic cell death happened directly and

**Fig. 3 | The PUFA-PL peroxidation based on antigen-directed binding between TIOs and cancer cells. a** The TIOs actively bind tumour cells and mediated PUFA-PLs peroxidation were shown schematically. In the recognition stage, the TIOs recognized the tumour cells by antigen coordination (e.g., VCAM-1 and integrins). In the peroxidation stage, the NIR light controlled ROS were released to the plasma membrane of the tumour cells and mediated the peroxidation reaction. **b** Confocal images of the antigens directed binding between catalytic amount of DiD-labelled TIOs (purple) and Calcein-AM-labelled tumour cells (green). Top, CT26 cells. Middle, 4T1 cells. Bottom, EMT6 cells. Scale bar, 50 μm (40×, 10×). **c** The NIR controlled cargo transportation from TIOs to 4T1 cells. The RSL3 were modified with rhodamine. The 4T1 cells were labelled with Calcein-AM (green). **c** Payloads transportation from TIOs to tumour cells. **d** Representative confocal images of the scope of the lipid peroxidation in TIOs bound 4T1 cells (NIR light, 980 nm, 0.2 W cm$^{-2}$, 1, 2 or 5 min). Scale bars, 10 μm. **e** The lipid peroxidation was detected by BODIPY 581/591

C11 (10 μM, 30 min incubation. Red, non-oxidized BODIPY; Green, oxidized BODIPY). **f** The NIR light induced lipid peroxidation with indicated time were analyzed by FACS. 4T1 cells were pretreated with BODIPY 581/591 C11. $n = 3$ independent experiments. Data are presented as mean ± s.e.m as appropriate (two-tailed unpaired Student's $t$-test was performed). **g** The MDA assay of TIOs bound 4T1 cells treated with NIR light, without NIR light or antioxidants. $n = 3$ independent experiments. Data are presented as mean ± s.e.m as appropriate (two-tailed unpaired Student's $t$-test was performed, $P < 0.0001$). **h, i** PUFA-PLs peroxidation analysis of 4T1 cells by mass spectrum. **h** The separated plasma membrane was analyzed using unsaturated degrees as scale. **i** The PUFA-PLs decrease were analyzed using unsaturated degrees as scale. For **b–d, h, i**, experiment was repeated three times independently with similar results. Source data are provided as a Source Data file.

quickly (Supplementary Movie 1–6), indicting the potent oxidative damages on the cell membrane. All the data above could verify our TIOs can break through the natural restrictions to mediate the lytic cell death.

Then, we investigated the killing mechanism of TIOs, as designed, the TIOs mediated cell lysis was performed by NIR light controlled PUFA-PLs peroxidation with the help of antigen-directed recognition. Obviously, both the lipid peroxidation and recognition were basic factors for TIOs mediated cell lysis (Fig. 4c). Thus, when enough antioxidants (e.g., Trolox, Ferrostatin-1, and Liproxstatin-1) were applied to NIR light treated TIOs bound tumour cells, significant decreases were found for both LDH and CCK8 (Fig. 4d and Supplementary Fig. 5a, b), while no influences were found in z-VAD inhibition assays (Fig. 4e and Supplementary Fig. 5c, d), indicating the cell death was mediated by lipid peroxidation instead of other caspase-dependent cell death mechanisms. Meanwhile, after irradiation, the downregulation of GPX4 in 4T1 cells were found in three independent experiments, which showed the inhibitory function of RSL3 (Fig. 4f and Supplementary Fig. 5e), as expected, the cell lysis effect disappeared when we took the RSL3 off the TIOs (Fig. 4g), which reconfirmed a successful lipid peroxidation reaction in living cells needs both ROS and RSL3. In addition, antigen-directed coordination played an important role in the TIOs-mediated cell lysis, because this not only gave TIOs the ability to kill cancer cells accurately, but also improves the lipid oxidation ability of ROS and RSL3 (Fig. 4c). Here, we employed a "Transwell" as a physical barrier for the TIOs and tumour cells to block the recognition between TIOs and tumour cells. After irradiation, as the produced ROS and RSL3 cannot be transported to the tumour cells directly, a significant drop in LDH release and a rise in cell viability were found (Fig. 4h), implying the significance of antigen-based recognition in the TIOs mediated cell death. All the data suggested the potent lytic cell death induced by TIOs is taking advantage of both antigen-directed recognition and ROS-mediated lipid peroxidation.

Notably, the intelligence of cytotoxic T cell presented not only in antigen-based precise attacking, but also in catalytic amount of T cells could eliminate the entire tumours[24]. In fact, the catalytic killing ability is crucial for any artificial anti-cancer drugs or systems, because the relative amount of them is very low at the focus. TIOs could mediate lytic cell death in a catalytic manner by continuous NIR light input. To illustrate this advantage, we first test the photostability of TIOs, under pulsed NIR light of different durations, the results showed the suspended TIOs were highly stable in 980 nm light and the ROS-producing was also sustainable (Supplementary Fig. 5f). Then, catalytic amount of TIOs were added to the 4T1 cells, after NIR light irradiation, the LDH assays showed the cell lysis could be continued by increasing the irradiation times (Fig. 4i and Supplementary Fig. 5g), indicating TIO system could perform killing tasks continually until all dangerous cells were eliminated.

## Tumour targeting of the TIOs in mice and light-controlled specific lipid peroxidation in tumours

As our TIOs exhibited good performance in light-controllable lytic cell death by PUFA-PL peroxidation, we explored its tumour targeting ability in mice. Although abundant evidence showed that engineered living macrophages could target tumours[35,36], but the targeting ability of TIOs was unknown. Magnetic particle imaging (MPI) has been widely used to characterize the biodistribution of cell-based drugs (e.g., CAR-T cells). Here, we used MPI-tracer $Fe_3O_4$ nanoparticles to label TIOs (TIOs-$Fe_3O_4$) and performed MPI imaging. 4T1, EMT6 and CT26 tumour-bearing mice were administered the TIOs-$Fe_3O_4$ individually by tail-vein injection. Time-dependent imaging and grey value analysis were carried out to track the biodistribution of the TIOs (Fig. 5a and Supplementary Figs. 6–8). The results showed that the TIOs accumulated in tumours after 6 h of circulation, reached extreme values within 24 h, and then decreased until 96 h (Fig. 5a and Supplementary Figs. 6–8). The accumulation of the TIOs in livers, lungs, spleens and kidneys were low relative to that in tumours (Fig. 5a and Supplementary Figs. 6a–b, d, 7a–c, e, 8a–c, e), and the average tumour-liver ratio was high (Fig. 5b, the maximum value was 4.5 at the 72-h time point). To exclude the interference that may came from the spontaneous leakage of $Fe_3O_4$ nanoparticles, we investigated the biodistribution of $Fe_3O_4$ nanoparticles themselves, the results showed they were mainly concentrated in livers, not tumours (Supplementary Figs. 6c, 7d, 8d)., These data showed that the TIOs presented good tumour targeting. Moreover, compare to the nanoscale RSL3@azo-Ce6-mSiO$_2$-UCNPs (Supplementary Fig. 9a, b), TIOs showed better tumour selectivity which was probably due to the liquid nitrogen treated-macrophage-based structure, as this treatment not only exposed more directing antigens, but also decreased immunogenicity of TIOs themselves (Supplementary Figs. 1j and 2a). Fluorescence staining of blood vessels at tumour sites showed a large amount of TIOs around the blood vessels, indicating the delivery and infiltration of TIOs were achieved through blood vessels (Supplementary Fig. 9c). Thus, although the TIOs is a micron-sized drug, tumour penetration was still good; for an 800 μm diameter tumour longitudinal section, the infiltration depth of the TIOs can reach 100 μm (Fig. 5c).

As NIR light presents spatiotemporal specificity, thus the NIR light controlled lipid peroxidation is tumour specific. For TIOs-treated 4T1 tumour-bearing mice, NIR light irradiation was performed at 48 h, and the treated mice were euthanized. Then, tumours were collected and sectioned, and strong oxidative signal was observed by fluorescence staining (Fig. 5d). The above tumours were prepared as a single-cell suspension for C11 BODIPY 581/591 labelling and FACS analysis, and the results showed high ROS and lipid peroxidation level in the NIR light treated tumours (Fig. 5d, e and Supplementary Fig. 9d). Moreover, the content of PUFA-PLs in tumour single-cell suspension was further analysed by mass spectrometry. Similar to the results in vitro, mass spectrometry showed kinds of PUFA-PLs decreased significantly after TIOs+NIR[1] treatment (Fig. 5f). The above data showed that the TIOs can

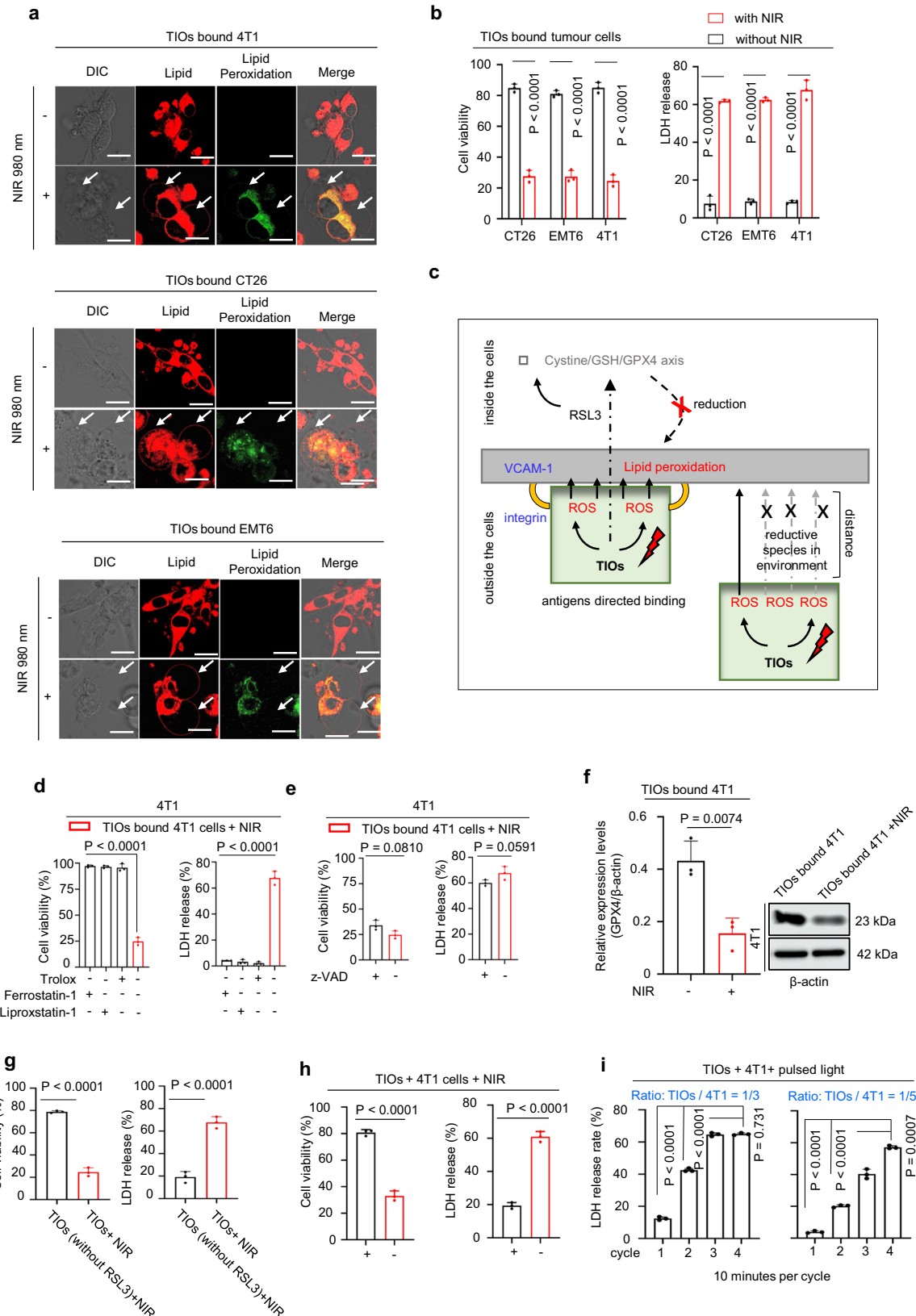

target tumours and perform tumour-specific light-controlled lipid peroxidation.

## The TIOs-mediated antitumour effects

As the TIOs was mainly enriched in tumour tissue and the NIR light controlled lipid peroxidation was tumour specific, we examined TIOs-

induced oncolysis and antitumour effects in tumour-bearing mice. Basing on the catalytic and controllable killing abilities of TIOs, we designed the fellow treatment schemes. 4T1 tumour-bearing mice (BALB/c mice) were injected intravenously with the TIOs on day 7, and then NIR light irradiation was carried out three times sequentially on days 8, 9 and 10 (Fig. 5g). In PBS-treated mice, the tumour volume

**Fig. 4 | TIOs-mediated lytic cell death in vitro. a** The fluorescence confocal microscopy of TIOs-mediated lytic cell death in 4T1, CT26 and EMT6 cells. NIR light (980 nm, 1.0 W cm$^{-2}$, 5 min). Scale bars, 10 μm. **b** The CCK-8 assays (left) and LDH release assays (right). The TIOs bound tumour cells were treated with or without NIR light (980 nm, 1.0 W cm$^{-2}$, 5 min). Red column, with NIR light. Black column, without NIR light. $n = 3$ independent experiments. $P < 0.0001$. **c** The diagram showed the mechanism of TIOs mediated cell lysis is based on ROS and RSL3 mediated lipid peroxidation, and the high efficiency of cell lysis is benefited from the antigen-directed recognition. **d, e** The CCK-8 and LDH release assays in inhibitor experiments. $n = 3$ independent experiments. **d** The TIOs bound 4T1 cells were irradiated by NIR light (980 nm, 1.0 W cm$^{-2}$, 5 min) with or without antioxidants as indicated. Antioxidants (e.g.,Trolox, Ferrostatin-1, Liproxstatin-1). $P < 0.0001$. **e** The

TIOs bound 4T1 cells were irradiated by NIR light (980 nm, 1.0 W cm$^{-2}$, 5 min) with or without z-VAD. **f** Statistical value of three independent samples and representative immunoblots for GPX4 in the indicated tumour cells. **g** The CCK-8 (left) and LDH release assays (right) of TIOs mediated cell death with or without RSL3 inside the TIOs. $n = 3$ independent experiments. $P < 0.0001$. **h** The transwell assay. $n = 3$ independent experiments. $P < 0.0001$. **i** The catalytic amount of TIOs mediated cell lysis by measuring the LDH release. $n = 3$ independent experiments. The pulsed NIR light (1.0 W cm$^{-2}$, 10 min per cycle) was used to treated the 4T1 cells. All data in **b**, **d–i** are shown as mean ± s.e.m (two-tailed unpaired Student's $t$-test was performed). For **a**, experiment was repeated three times independently with similar results. Source data are provided as a Source Data file.

rapidly increased to 20-fold higher than the initial value in 2 weeks (Fig. 5h). In TIOs+NIR[3]-treated mice, apparent inflammatory symptoms were observed in tumour tissues after irradiation, and massive tumour shrinkage occurred during the following 15 days (Fig. 5h). On day 25, a minimal tumour burden could be detected (Fig. 5j). In contrast, mice treated with the TIOs alone or RSL3@azo-Ce6-mSiO$_2$-UCNPs+NIR[3] did not show any antitumour effects (Fig. 5j–h, k). Real-time temperature monitoring of the tumour revealed that the thermal effect was insignificant during irradiation, indicating the antitumour effects came from oncolysis (Supplementary Fig. 10a). Survival curves showed a benefit after TIOs+NIR[3]-treatment (Fig. 5l). Notably, intratumoral injection of ferrostatin-1 abolished the antitumour effects of TIOs +NIR[3], which suggested pronounced tumour regression resulting from lipid peroxidation (Fig. 5j–h, k). All mice in the TIOs+NIR[3] treatment group maintained normal body weight, low IL-6 release, and body temperature (Fig. 5i and Supplementary Fig. 10b–e), and the tissue sections showed negligible damage to the major organs after oncolysis (Supplementary Fig. 11), indicating TIOs+NIR[3] mediated-oncolysis was safe. Similar antitumour effects could be observed in CT26 or EMT6 tumour-bearing mice, which suggests that the TIOs-mediated anti-tumour effects are not limited to a particular tumour model (Supplementary Fig. 12). Notably, all the 4T1, CT26 and EMT6 tumours were PD-L1 positive (Supplementary Fig. 4c), thus the above data showed our TIOs had a better tumour killing ability than natural CD8$^+$ T cells in terms of breaking through immune checkpoint inhibition.

**Sufficient oncolysis induced the potent antitumour immunity and immune memory**

However, there was no tumour shrinkage when we repeated the above treatments in immunodeficient mice, even the lipid peroxidation still be found in TIOs+NIR[3]-treated immunodeficient mice, indicating that the TIOs triggered tumour regression was mediated by the antitumour immune function (Fig. 6a, b and Supplementary Fig. 13a–c, d–f). A detailed T-cell flow cytometry analysis was performed in BALB/c mice engrafted subcutaneously with 4T1 tumour cells, and a drastically larger tumour CD3$^+$ T-cell population was observed in TIOs+NIR[3]-treated mice than in tumours from PBS-treated mice and TIOs alone-treated mice (Fig. 6l). Among the T cells, both the CD4$^+$ and CD8$^+$ subpopulations were ~10-fold larger, while the percentage of CD4$^+$Foxp3$^+$ T regulatory (Treg) cells, which are negative regulators of antitumour immunity, was much smaller in tumours from TIOs+NIR[3]-treated mice (Fig. 6l). After functional depletion of CD4$^+$ T and CD8$^+$ T cells in 4T1 tumour-bearing BALB/c mice, the antitumour effects disappeared in TIOs+NIR[3]-treated mice (Fig. 6c and Supplementary Fig. 13g–i). These data clearly indicated that the function of CD4$^+$ helper T cells and cytotoxic T cells was activated by TIOs mediated lytic cell death in tumours[37].

To obtain a full picture of TIOs-induced immunological changes in the tumours, CD45$^+$ leucocytes from 4T1 tumour-bearing PBS-treated and TIOs+NIR[3]-treated mice were subjected to single-cell RNA sequencing (scRNA-Seq) analyses. A total of 15603 single immune cells were analysed and clustered into 9 subsets on the two-dimensional t-

SNE map (Fig. 6d). For TIOs+NIR[3]-treated mice, increased populations of CD4$^+$, CD8$^+$, NK (natural killer), M1 macrophages, and monocyte cells but decreased percentages of M2 macrophages and neutrophils were observed. Notably, the larger subpopulation of M1 macrophages originated from M2 macrophages, because expression of the M2 marker (Arg1) was decreased (Fig. 6e and Supplementary Figs. 14–15). Further analyses of the expression data revealed that genes encoding chemotactic cytokines such as Ccl5, Cxcl9 and Cxcl10 as well as genes important for T and/or NK-cell activation (Cd69 and Klrk1) had elevated expression in TIOs+NIR[3]-treated mice (Supplementary Fig. 14). In addition, the expression of effector genes critical for antitumour immunity (such as Ifng, Gzms and Fasl) was also upregulated in the therapeutic tumour samples (Supplementary Fig. 14).

Metastasis is a major challenge in the cancer treatments[1,2]. For TIOs+NIR[3] treated mice, no metastases appeared at day 25, while lung metastases were found in PBS-treated and TIOs alone treated mice (Supplementary Fig. 16). Encouraged by the TIOs-induced antitumour immune response, we explored immune memory and vaccination effect in BALB/c mice engrafted subcutaneously with 4T1 tumour cells. On day 11, 4T1 cells were subcutaneously injected again into the left sides of mice that received complete TIOs+NIR[3] treatments or PBS treatment (Supplementary Fig. 17). Encouragingly, the mice treated with TIOs+NIR[3] had the ability to resist the second transplanted tumours for 50 days, while the tumours grew rapidly on both sides of PBS-treated mice (Supplementary Fig. 17).

As in the scRNA-Seq analyses, the population of monocytes and M1 macrophages increased significantly in TIOs+NIR[3] treated 4T1 tumour-bearing BALB/c mice (Supplementary Fig. 15c), which indicated a typical inflammatory symptom[38]. We assessed the concentrations of proinflammatory factors (IL-1β, IL-18, HMGB1) in tumours or in serum. The results showed that in TIOs+NIR[3] treated 4T1 tumour-bearing BALB/c mice, the level of inflammation was hundreds-fold higher in the tumour bed, indicating an acute inflammation has been triggered in tumours (Supplementary Fig. 18a). Compared with that in tumours, the increase of IL-1β, IL-18, and HMGB1 levels in the serum was not severe, suggesting that TIOs+NIR[3]-triggered antitumour immune response is relatively safe and does not cause hyperactive immune response or substantial systematic inflammation (Supplementary Fig. 18b). Notably, in the above acute inflammatory environments, genes that are known to have protumour or immunosuppressive effects, such as Csf1, Vegfa, Arg1, Cd274 (encoding PD-1) and Pdcd1lg2 (encoding PD-L2), showed decreased expression upon TIOs+NIR[3] treatment (Fig. 6b and Supplementary Fig. 15b), suggesting the acute inflammation could benefit the potent anti-tumour immune function by not only increasing the infiltration of T cells but also improving the vitality of T cells.

**STING activation rescued the low-intense oncolysis caused T-cell exhaustion**

As mentioned, excessive lytic cell death may lead to cytokine release syndrome (CRS). Therefore, from the perspective of safety, the induction of antitumor immunity through low intense oncolysis has

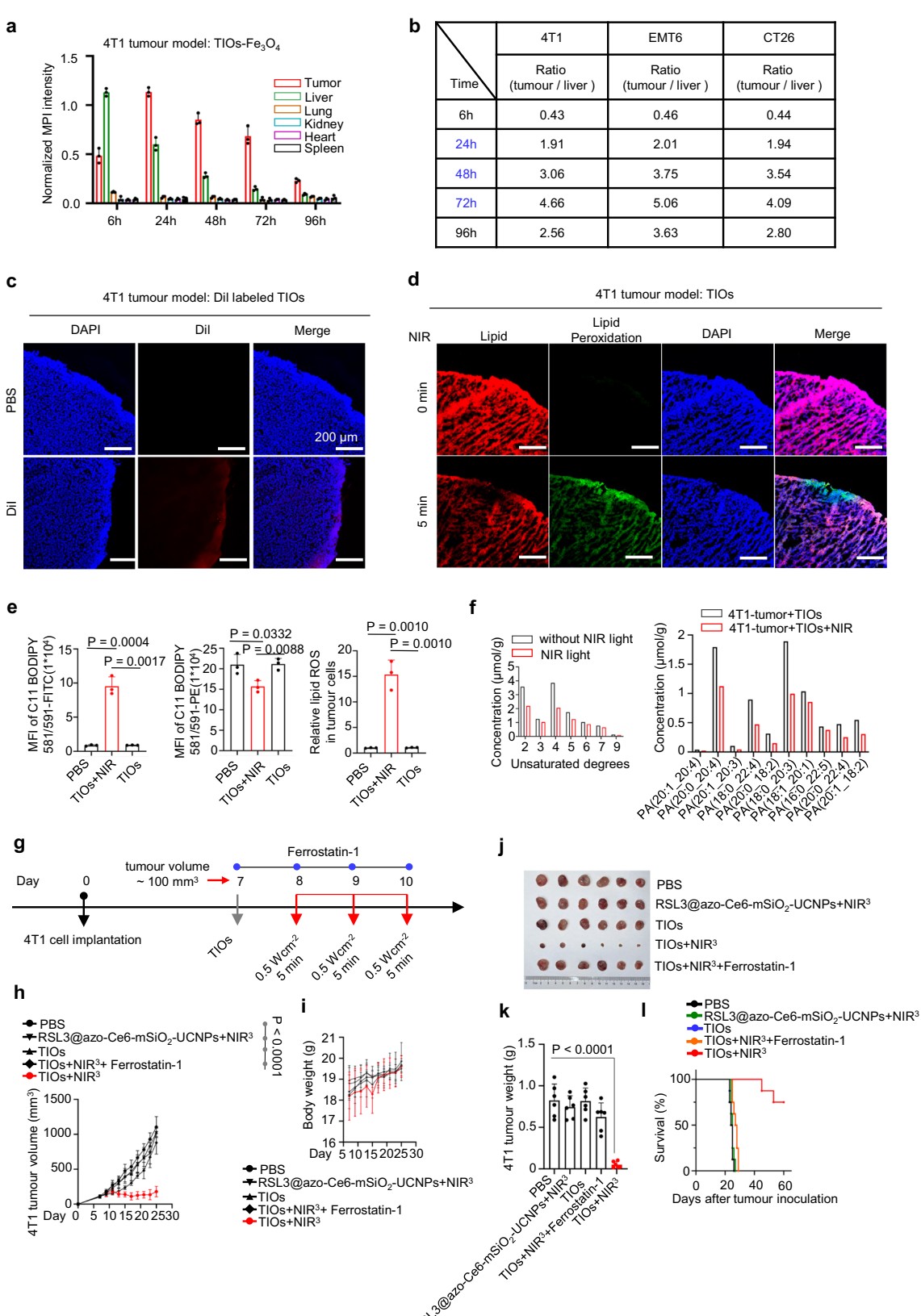

more clinical application prospects[27]. Fortunately, the TIO system was a reliable tool for tuning the efficiency of oncolysis or the intensity of cytokine release by NIR light. Thus, we investigated whether a reduced TIOs mediated oncolysis could also trigger antitumour immune function. Same amount of TIOs were intravenously injected into 4T1 tumour bearing mice, then a single time NIR light irradiation was used.

The T-cell flow cytometry analysis showed, in TIOs+NIR[1] treated tumour beds, the population of CD3[+], CD4[+] and CD8[+] T cells could be increased to the same level of TIOs+NIR[3] treated mice, and the Treg cells also decreased (Fig. 6m). All the above data pointed a substantial immune response after reduced oncolysis and gave us a confidence in reduced oncolysis mediated antitumour immunotherapy. However,

**Fig. 5 | In vivo metabolic performance of TIOs and TIOs induced lipid oxidation and antitumour effect. a** Average intensity of $Fe_3O_4$-labelled TIOs ($5 \times 106$ units / mouse) in the corresponding organs of 4T1 tumour models at indicated time points. $n = 3$ mice. **b** The average intensity ratio of tumour to liver at indicated time points. $n = 3$ mice. **c** Representative confocal images of tumour sections show the penetration of Dil-labelled TIOs in 4T1 tumour-bearing mice after intravenous injection. Injection dose, $5 \times 106$ units/mouse. Time point, 24 h after injection. **d** Representative fluorescence images of lipid peroxidation based on BODIPY® 581/591 C11 stained 4T1 tumours. TIOs+NIR treated mice (980 nm, $0.5\ W\ cm^{-2}$, 5 min) or TIOs alone treated mice. Scale bars, 10 µm. **e** Tumour lipid peroxidation of 4T1 tumour-bearing mice treated with PBS, TIOs+NIR (980 nm, $0.5\ W\ cm^{-2}$, 5 min) or TIOs alone, $n = 3$ mice for indicated groups. **f** Mass spectrometric analysis of TIOs induced lipid peroxidation in 4T1 tumours with or without NIR light. **g–l** Analysis of the antitumor effects in 4T1 tumour-bearing mice treated with TIOs+NIR[3]. TIOs +NIR[3] treatment scheme in BALB/c mice implanted subcutaneously with 4T1 cells ($n = 6$ mice per group). **g** The tumour-bearing mice (average tumour volume: 100 $mm^3$) were intravenously injected (*iv*) with TIOs (*iv*, $5 \times 106$ units/mouse) and then treated with NIR light or not (980 nm, $0.5\ W\ cm^{-2}$, 5 min). Ferrostatin-1 was injected intratumorally on days 7, 8, 9, and 10. (it, $5.0\ mg\ kg^{-1}$ per day). **h** Average tumour volume. $P < 0.0001$. **i** Average mice weight. **j** Photographs of representative tumours on day 25. **k** Average tumour weight on day 25. $P < 0.0001$. **l** Survival curves of 4T1 tumour-bearing mice. **a**, **b**, **e**, **h**, **l**, **k**, s.e.m. data are shown and two-tailed unpaired Student's *t*-test was performed. Data shown are representative of two (**a**, **b**, **e**, **f**, **h–l**) or three (**c**, **d**) independent experiments. Source data are provided as a Source Data file.

the tumour volume growth rapidly in TIOs+NIR[1] treated mice (Fig. 6g), indicating though the reduced oncolysis could recruit enough T cells to infiltrate the tumours, no antitumor effects were generated. The contradiction between therapeutic effects and immune response urged us to explore the reciprocal loops of cell death, inflammation and immunity in reduced oncolysis.

A more detailed immune analysis showed the effect memeroy−like subset ($T_{EM}$, CD62L[+] CD44[−]) of CD8[+] T cells[15] increased in TIOs+NIR[1] treated mice (Fig. 6j), indicating a chronic inflammatory environment in TIOs+NIR[1] treated tumours[39]. Recently, emerging evidence showed the T-cell exhaustion happened in chronic inflammatory tumour microenvironment and led to immune dysfunction[15,40]. Inspired by this point, we analysed the inflammation in TIOs+NIR[1] treated mice, as shown, compared with TIOs+NIR[3] treatment, the IL-1β, IL-18 and HMGB-1 release in tumour beds were almost four fifths lower (Supplementary Fig. 17a, c). Notably, although the level of inflammation was low, it lasted a long time (Supplementary Figs. 17, 23). As reported early, in a chronic inflammatory environment, the CD8[+] T cell would be exhausted by upregulating the PD-1 expression[15,40,41]. Thus, we analysed the PD-1 expression of tumour infiltrated CD8[+] T cells in TIOs+NIR[1] treated 4T1 tumour bearing mice by flow cytometry. Different with the down-regulated PD-1 expression in TIOs+NIR[3] treatment, after NIR light irradiation, the PD-1 expression increased in TIOs +NIR[1] treated mice (Fig. 6j). Thus, we investigated whether synergizing with anti-PD-1 therapy could rescue the failure in TIOs+NIR[1] treatments[41]. During the four-times anti-PD-1 treatments, the tumour regression occurred in TIOs+NIR[1] + PD-1-treated mice, which revealed that the reduced oncolysis-induced antitumour immunity could be enhanced by anti-PD-1 therapy (Fig. 6f–h and Supplementary Fig. 19a–c). However, after finishing the administration of anti-PD-1 antibodies, the tumour volumes began to rebound again and quickly reached 1000 $mm^3$ at day 35 (Fig. 6h). In fact, this phenomenon was reasonable, because anti-PD-1 antibody could only relieve the killing inhibition of tumour infiltrating CD8[+] T cells but could not fundamentally reshape the exhausted T cells[40,41]. Therefore, after anti-PD-1 antibodies were completely metabolized in the mice, the tumours would show immune dysfunction again[41]. The expression of TCF1, a transcription factor marker of stem cell−like T cells has been used to indicate the stemness of PD-1[+]CD8[+] T cells[40,41]. In TIOs+NIR1 treated mice, the intensity of TCF1 decreased, showing the stemness of PD-1[+] CD8[+]T cells were low. The cGAS-STING have showed a positive role in the differentiation of stem cell−like T cells, thus manipulating cGAS-STING pathway may enhance T cell-based immunotherapy[15,40,42]. In TIOs+NIR[1] + cGAMP[4] treated 4T1 tumour bearing mice, four times cGAMP were injected intratumorally (Fig. 5f), compared with PBS-treated, cGAMP alone treated and TIOs+NIR[1] treated mice, the tumour disappeared by treating with TIOs+NIR[1] + cGAMP[4] (Fig. 6i). Notably, the impressive tumour immune response and the high benefits of combined therapy can maintain in 50 days (Fig. 6i), suggesting STING activation released a more powerful boost to durative antitumour immunity. Then, we performed a detailed T cell analysis, the results showed, after STING activation, the PD-1 expression was decreased in CD8[+] T cells, moreover, the $T_{EM}$ subset (CD62L[+] CD44[−]) of CD8[+] T cells and the stemness of PD-1 positive CD8[+] T cells was increased (Fig. 6k), indicating STING activation reshaped tumour-infiltrating T cells[15].

## Discussion
Here, we showed a T-cell-inspired oncolytic system efficiently triggered lytic cell death in a NIR light controllable and catalytic manner, which provided insight for designing artificial systems to break through the natural restrictions to manipulate oncolysis. The application of our system in vivo confirmed the powerful antitumour immune function and immune double-edged sword effect in oncolysis. Thus, a CAR-T cells like system capable of targeting peroxidised PUFA-PLs may improve the efficacy of anti-cancer immunotherapy. An artificial oncolytic system independent of intrinsic mechanisms provided an approach for understanding the unique nature of immunogenicity of oncolysis induced antitumor immunity. The initial intensity of inflammation induced by oncolysis was critical for the generation of potent antitumour immunity and immune memory. The chronic inflammation caused by oncolysis may result in T-cell exhaustion and tumour progression. The synergising oncolysis with STING activator could provide a persistent stimulus for antitumor immunity by reshaping the stemness of CD8[+] T cells.

## Methods
### Data reporting
No statistical methods were used to predetermine sample size. The experiments were not randomized and investigators were not blinded to allocation during experiments and outcome assessment. Ethical compliance with the IACUC protocol was maintained. In none of the experiments did the size of the tumour graft surpass 2 cm in any two dimensions, and no mouse had severe abdominal distension (≥10% increase in original body weight), as outlined by the Animal Experimental Ethical Inspection committee of the Laboratory Animal Centre, Wenzhou Medical University (ID number: xmsq2021-0530).

### Materials
Annexin V-FITC and propidium iodide staining kits (556547) were purchased from BD Biosciences. Singlet Oxygen Sensor Green Reagent (S36002), Image-iT® Lipid Peroxidation Kit (C10445), NucBlue™ Live ReadyProbes™ Reagent (Hoechst 33342) and ProLong™ Diamond Antifade Mountant with DAPI (P36971) were obtained from Invitrogen. PageRuler™ Prestained Protein Ladder (26617) was obtained from Thermo Scientific. DiD Perchlorate (DiIC18(5)) for live-cell imaging (40758ES25) were obtained from Yeasen Biotechnology. Ferrostatin-1 (S7243) and liproxstatin-1 (S7699) were purchased from Selleck. Calcein-AM and Trolox were purchased from Sigma–Aldrich. Plasma Membrane Protein Isolation and Cell Isolation Kit (Minute™, SM-005) was purchased from Invent Biotechnologies. Lipid Peroxidation MDA Assay Kit (S0131S) was purchased from Invent Biotechnologies Beyotime. Cell viability and LDH release assays were performed by using the

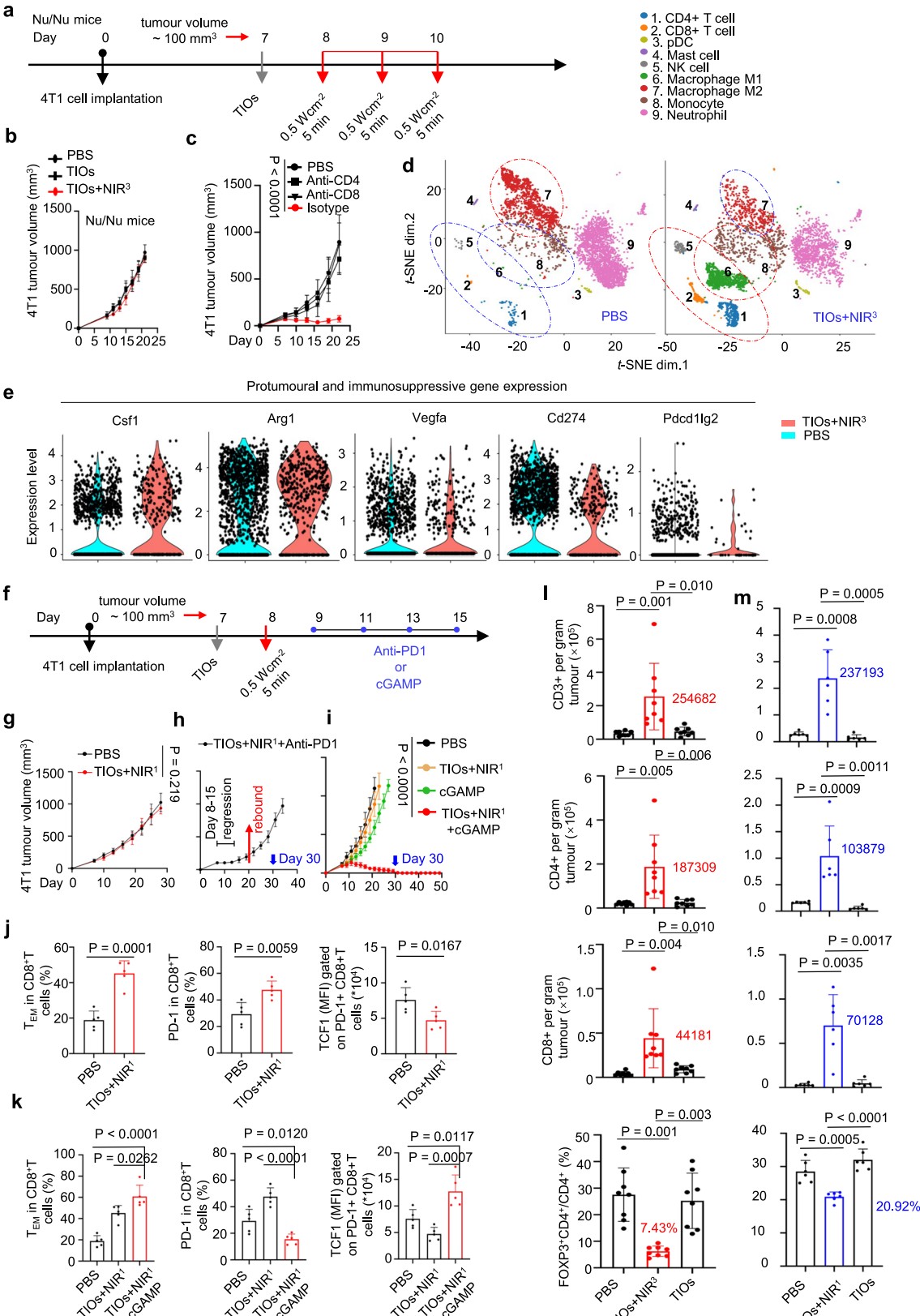

Cell Counting Kit-8 (DOJINDO, CK04-20) and Cytotoxicity LDH Assay (DOJINDO, 341-91754), respectively. Enzyme-linked immunosorbent assay (ELISA) kits for IL-1β, IL-6, IL-18, and HMGB1 were purchased from MEIMIAN Biology.

Anti-glutathione peroxidase 4 (ab125066), integrin α4 (D2E1) XP® rabbit mAb (CST8440), integrin β1 (D6S1 W) rabbit mAb (CST34971), VCAM-1 (D8U5 V) rabbit mAb (CST39036), SIRPα/SHPS1 (D6I3 M) rabbit mAb (CST13379), anti-rabbit IgG (H + L), F(ab')2 fragment (Alexa Fluor® 488 conjugate) (CST4412), anti-β-actin (CST8457) were purchased from Cell Signalling Technology. Anti-GSDME (ab215191) and anti-GSDMD (ab209845 or ab219800) were purchased from Abcam. CD24 monoclonal antibody (M1/69) (14-0242-82) was purchased from

**Fig. 6 | Oncolysis induced antitumour immune function and the immune double-edged sword effect. a, b** Evaluation of the efficacy of TIOs induced antitumor effects in Nu/Nu mice. *n* = 6 mice per group. **a** TIOs+NIR[3] treatment scheme in Nu/Nu mice bearing 4T1 tumours (average tumour volume: 100 mm³) were treated by PBS, TIOs treated with NIR light or not (980 nm, 0.5 W cm⁻², 5 min). **b** Average tumour volume of TIOs+NIR[3] treatment in tumour bearing Nu/Nu mice. **c** Average tumour volume of TIOs+NIR[3] treatment in T cell depletion assays. Depletion of the CD4⁺ T or CD8⁺ T cell population blocked 4T1 tumour regression induced by TIOs+NIR[3]. *n* = 6 mice. *P* < 0.0001. **d** Single-cell RNA sequencing of CD45⁺ immune cells from 4T1 tumours treated with PBS or TIOs+NIR[3]. t-distributed stochastic neighbour embedding (t-SNE) plots of CD45⁺ cells randomly sampled from each group are shown. **e** Protumoural and immunosuppressive gene expression in PBS or TIOs+NIR[3] treated mice. **f, i** Antitumour immunotherapy of TIOs+NIR[1]-mediated oncolysis synergized with anti-PD-1 antibody or cGAMP. **f** The treatment scheme (*n* = 6 mice for indicated groups, average tumour volume: 100 mm³). **g** Average tumour volume of PBS treated mice or TIOs+NIR[1]-treated mice. **h** Average tumour volume of TIOs+NIR[1]-treated mice synergised with anti-PD-1 therapy. **i** Average tumour volume of indicated treatment. *P* < 0.0001. **j, k** The analysis of CD8⁺ T-cell exhaustion based on central memory–like subset (CD62L⁺ CD44⁺), PD-1 expression and stemness. *n* = 5 mice per group. **j** The CD8⁺ T cells in PBS and TIOs+NIR[1] treated mice. **k** The CD8⁺ T cells in PBS, TIOs⁺NIR[1] and TIOs +NIR[1] + cGAMP treated mice. **l, m** The CD3⁺, CD4⁺, CD8⁺ T cells and Treg cells in mice were analysed by FACS. **l** The mice were treated with PBS, TIOs+NIR[3] and TIOs alone. *n* = 8 mice per group. **m** The mice were treated with PBS, TIOs+NIR[3] and TIOs alone. *n* = 6 mice per group. **b, c**, and **g–m**, Data (mean ± s.e.m.) were analyzed by two-tailed unpaired Student's *t*-test. Data are representative of two (**b–d, g–m**) independent experiments.Source data are provided as a Source Data file.

Invitrogen. The CD47 rabbit mAb (A1838) was purchased from ABclonal. Purified anti-mouse CD31 antibody (102402), Purified anti-mouse Siglec-G antibody (163302), Brilliant Violet 785™ anti-mouse CD274 (B7-H1, PD-L1) antibody was purchased from BioLegend. For fluorescence-activated cell sorting (FACS) analyses of tumour-infiltrating lymphocytes, FITC anti-mouse CD45 antibody-S18009F, APC/Cyanine7 anti-mouse CD3-17A2, APC anti-mouse CD4-RM4-4, and PE/Cyanine7 anti-mouse CD8a-53–6.7 were purchased from BioLegend. The PE FOXP3 monoclonal antibody (FJK-16s) was obtained from Invitrogen. CD44-PerCP-Cy5.5 (IM7), CD62L-BV510 (MEL-14) were purchased from BD Biosciences. PD-1-BV605 (J43) was purchased from BioLegend.

The InVivoMAb anti-mouse PD-1 antibody (clone J43) used to treat 4T1 tumours was obtained from BioXcell. For immune cell depletion, anti-mouse CD4 (clone GK1.5), anti-mouse CD8 (clone YTS 169.4), anti-mouse IL-1β (clone B122) and anti-mouse IL-18 (clone YIGIF74-1G7) antibodies were produced by BioXcell.

### Cell lines and cell culture conditions

4T1, CT26 and Raw 264.7 cells were kindly provided by the Stem Cell Bank, Chinese Academy of Sciences. EMT6, TC-1, AML12 and TCMK-1 cells were obtained from the American Type Culture Collection (ATCC). Raw 264.7 cells were grown in Dulbecco's modified Eagle's medium (DMEM, Gibco) supplemented with 10% (v/v) foetal bovine serum (FBS, Gibco) and 1% Pen Strep (Gibco). 4T1, CT26, EMT6, TC-1, and cells were grown in RPMI 1640 medium (Gibco) containing 10% FBS and 1% Pen Strep. AML12 cells were grown in 1:1 mixture of Dulbecco's modified Eagle's medium and Ham's F12 medium with 0.005 mg/mL insulin, 0.005 mg/mL transferrin, 5 ng/mL selenium, and 40 ng/mL dexamethasone, 90%; foetal bovine serum, 10%. TCMK-1 cells were grown in the base medium for this cell line is ATCC-formulated Eagle's Minimum Essential Medium, Catalogue No. 30-2003. To make the complete growth medium, add the following components to the base medium: foetal bovine serum to a final concentration of 10%. All cells were grown at 37 °C with 5% CO₂. All cell lines were confirmed to be mycoplasma negative by the standard PCR method. The identity of the cells was frequently checked by morphological features but was not authenticated by short tandem repeat (STR) profiling. Mouse BMDMs were differentiated in vitro from isolated bone marrow cells. Bone marrow cells collected from mouse femurs and tibias were incubated for 7 days in DMEM containing 20% FBS, 1% Pen Strep, and 10 ng/mL M-CSF (Sigma–Aldrich).

### Preparation and characterization of the TIOs

**Preparation of TIOs.** The synthesis of RSL3@azo-Ce6-mSiO₂-UCNPs is described in the Supplementary Methods (see Supplementary Information). Then, 300 µg ml⁻¹ RSL3@azo-Ce6-mSiO₂-UCNPs were incubated with Raw 264.7 cells (-10⁷ cells/dish) for 6 h, followed by gentle washing with PBS or DMEM complete medium to generate the precursor of TIOs. the precursor of TIOs (-10⁷ cells/tube, 1.0 ml) were

immersed in liquid nitrogen, and after 24 h, they were thawed at 37 °C, centrifuged at 1000 rpm for 3 min, washed and resuspended in DMEM complete medium or PBS to generate TIOs (5 × 106 units/tube, 1.0 ml). The procedure used to generate RSL3-RhB load TIOs or BMDM version TIOs was similar to that used for TIOs.

**Characterization of the TIOs system.** The morphology, size distribution and zeta potential of the nanoparticles or TIOs were characterized by scanning electron microscopy (SEM, FEI Teneo), transmission electron microscope (TEM, FEI Tecnai Spirit) and dynamic light scattering (DLS, Malvern ZetasizerNano S90). The amount of loaded Ce6 was determined by the difference between the initial amount of Ce6 and the amount of Ce6 in the supernatant after centrifugation. The absorbance of Ce6 was determined by UV-visible spectrophotometer and quantified by the standard curve. The loading of azo and RSL3 in RSL3@azo-Ce6-mSiO₂-UCNPs was also determined using the same method as above.

**Singlet oxygen generation by TIOs.** After cotreatment with TIOs (10⁵ units/well) and singlet oxygen sensor green (SOSG) reagent (1.0 µl, 5.0 mM) in a 48-well plate for 5 min, followed by NIR light (980 nm, 1.0 W cm⁻²) for different time (2, 4, 6, 12, 24, 36, 60, 120, 180, 240, 300, 360, 600, 1200 and 1800 s), singlet oxygen was quantified by detecting the fluorescence of singlet oxygen (excitation/emission maxima ~504/525 nm) with a multimode reader (Spark, TECAN) and flow cytometry. Furthermore, TIOs were seeded in glass-bottom dishes (10⁶ units/dish) and coincubated with SOSG reagent (1.0 µl, 5.0 mM) for 5 min, followed by NIR light (980 nm, 1.0 W cm⁻²) for different time (5, 10, 15, and 20 min). The irradiation process was designed with an interval of 30 min after every 5 min of irradiation. Four cycles of irradiation were carried out before static image capture and flow cytometry. The fluorescence data for RhB and SOSG statistics were obtained from confocal images and analysis with Olympus FV31S software and flow cytometry.

**Stability of TIOs under physiological conditions.** To examine the stability of TIOs under physiological conditions, TIOs were resuspended in 1.0 ml PBS at 37 °C, and dynamic light scattering (DLS, Malvern Zetasizer Nano S90) combined with a cell counter was used at specified times to assess the change in TIOs. Stability at room temperature and 4 °C was also determined as described above.

### Membrane proteins evaluation of TIOs

**Flow cytometry.** The expression of SIRPα/SHPS1, CD47, Siglec-G, CD24, Integrin β1, Integrin α4 in the TIOs or BMDMs version TIOs, and the expression of VCAM-1 in tumour cells (EMT6, CT26, 4T1) were analysed by flow cytometry. Dispersed TIOs or BMDMs version TIOs (1 × 10⁶ units) in 100 µl of cell staining buffer, and add 0.25 µg of the corresponding antibodies (SIRPα/SHPS1 (D6I3 M) rabbit mAb, CD47 rabbit mAb, purified anti-mouse Siglec-G antibody, CD24 monoclonal

antibody (M1/69), integrin α4 (D2E1) XP® rabbit mAb, integrin β1 (D6S1 W) rabbit mAb), incubating at 4 °C for 20 min in the dark, washing twice with 2.0 ml cell staining buffer. Then, adding 100 µl anti-rabbit IgG (H + L), F(ab')2 fragment (Alexa Fluor® 488 conjugate) (1:1000) conjugated to the purified antibodies and incubate at 4 °C for 30 min in the dark. Meanwhile, dispersed EMT6, 4T1 and CT26 cells ($1 \times 10^6$) in 100 µl of cell staining buffer, then add 0.25 µg of the corresponding flow antibody (anti-VCAM-1-purified or Brilliant Violet 785™ anti-mouse CD274) and incubate at 4 °C for 20 min in the dark. Washing twice with 2 ml of cell staining buffer, adding 100 µl anti-rabbit IgG (H + L), F(ab')2 fragment (Alexa Fluor® 488 conjugate) (1:1000) and incubate at 4 °C for 30 min in the dark. Cells were washed twice with 2 ml of cell staining buffer and analysed by flow cytometry using Agilent Novocyte.

**Mass spectrometry.** Rapid isolation of native membrane proteins using the Plasma Membrane Protein Isolation and Cell Isolation Kit (Minute™) enables evaluation of membrane proteins on Raw 264.7 and TIOs. $5 \times 10^7$ cells were resuspended in lysis buffer (8.0 M urea, 1% protease inhibitor cocktail) and sonicated three times at 4 °C using a high-intensity sonicator (Scientz). Removing the debris by centrifugation at 12,000 g for 10 min at 4 °C. After that, the supernatants were collected and the membrane protein solution was obtained by gradient centrifugation according to the manufacturer's instructions. Finally, membrane proteins were identified by liquid chromatography tandem mass spectrometry analysis and the resulting LC-MS/MS data were processed using the MaxQuant search engine (v.1.6.15.0).

**In vitro tumour cell recognition and payload transportation**
**Evaluation of TIOs recognition of tumour cells.** Tumour cells were seeded in glass-bottom dishes (4T1, CT26 and EMT6, $10^6$ cells/dish) with a Calcein-AM ($10.0$ µL, $1.0$ mg ml$^{-1}$) containing cell culture medium for 6 h. The tumour cells were washed twice with PBS to generate Calcein-AM-labelled 4T1, EMT6 or CT26 cells. DiD-labelled aGranules$^{ferro}$-BK$^{raw}$ ($5 \times 10^6$ units/dish) were added and coincubated for 6 h, and unbound TIOs were removed by washing with PBS. Representative images of TIOs-bound tumour cells were observed and captured with an Olympus confocal microscope (Olympus FV3000). Binding of other cell lines (e.g., BMDC, BMDM, TC-1, AML12 and TCMK-1) to TIOs can also be treated according to this protocol.

**Payload delivery by TIOs in vitro.** TIOs-bound 4T1 cells were generated as described above. TIOs-bound 4T1 cells were seeded in glass-bottom dishes (-$10^6$ 4T1 cells/dish) and treated with NIR light (980 nm, $1.0$ W cm$^{-2}$). The irradiation process was designed with an interval of 30 min after every 5 min of irradiation. Four cycles of irradiation were carried out before static image capture and flow cytometry. The fluorescence data for RhB statistics were obtained from confocal images and analysis with Olympus FV31S software and flow cytometry with an Agilent Novocyte flow cytometer.

**Cell death assays**
**Microscopy imaging of cell death.** To observe the cell morphology after different treatments, tumour cells were seeded in glass-bottom dishes ($10^6$ cells/dish) for 6 h, exposed to TIOs for 6 h ($10^7$ units/dish), washed twice with PBS, and then treated with NIR light (980 nm, $1.0$ W cm$^{-2}$, 5 min). To perform lipid peroxidation staining, cells were resuspended in $1.0$ ml DMEM containing $10.0$ µM BODIPY 581/591 C11 and incubated for 30 min at 37 °C in a 5% $CO_2$ incubator. The cells were washed and resuspended in 500 µl fresh PBS (Gibco) and then analysed immediately on an Olympus confocal microscope. The image data shown are representative of at least three randomly selected fields.

**Lipid peroxidation assessed by BODIPY 581/591 C11 reagent.** 4T1 cells were seeded in glass-bottom dishes ($10^6$ cells/dish) for 6 h and add $1.0$ ml of RPMI 1640 medium containing 10 µM BODIPY 581/591 C11 and incubate for 30 min at 37 °C for lipid peroxidation staining. Remove media and wash cells three times with PBS. TIOs ($5 \times 10^6$ units/dish) were added and coincubated for 6 h and unbound TIOs were removed by washing with PBS, and then treated with NIR light (980 nm, $0.2$ W cm$^{-2}$) for different times (0, 1, 2 min). The fluorescence data for nonoxidized C11 (PE channel) and oxidized C11 (FITC channel) were obtained from confocal images and analysis with Olympus FV31S software and flow cytometry with an Agilent Novocyte flow cytometer. Determine lipid peroxidation by quantitating the fluorescence intensities and calculating the ratio of intensity in FITC channel to the intensity in PE channel. *Lipid peroxidation assessed by MDA assay.* TIOs-bound tumour cells were seeded into 6-well plates for 6 h and treated with NIR light (980 nm, $1.0$ W cm$^{-2}$, 5 min). The MDA content in tumour cells and tissues was measured using a Lipid Peroxidation MDA Assay Kit. Cells were lysed by adding 100 µl of cell lysis buffer per 1 million cells. After lysis, 100 µl of supernatant was obtained by centrifugation at 10,000 g–12,000 g for 10 min at 4 °C, followed by the addition of 200 µl of MDA assay working solution. After mixing, the mixture was heated at 100 °C for 15 min. The water bath was cooled to room temperature and centrifuged at 1000 g for 10 min at room temperature. Then, 200 µl of supernatant was added to a 96-well plate, and the absorbance was measured at 532 nm with a microplate reader.

**Cell viability and LDH release assays.** TIOs-treated tumour cells were cultured in a 48-well plate ($10^4$ cells/well) for 6 h before being subjected to NIR light (980 nm, $1.0$ W cm$^{-2}$, 5 min). The pan-caspase inhibitor Z-VAD-FMK (z-VAD, $50.0$ µM, 24 h), the lipid peroxidation inhibitors ferrostatin-1 (Fer-1, $2.0$ µM, 24 h), liproxstatin-1 ($1.0$ µM, 24 h) and Trolox ($1.0$ µM, 24 h) were added to the culture medium respectively to detect effects on LDH release, cell viability and MDA content. In addition, the equivalent number of RSL3@azo-Ce6-mSiO₂-UCNPs-treated but not TIOs-treated 4T1 cells were cultured in a 48-well plate for 6 h before being exposed to NIR light (980 nm, $1.0$ W cm$^{-2}$, 5 min). The pan-caspase inhibitor Z-VAD-FMK (z-VAD, $50.0$ µM, 24 h) was added to the culture medium to detect effects on LDH release and cell viability. The experiments were conducted by following vendor-provided protocols, and all tests were repeated at least three times.

**Transwell assay.** The 4T1 cells ($10^4$ cells/well) were cultured in the bottom wells, and the TIOs ($10^5$ units/well) were cultured in the top wells. The samples were treated with or without NIR light (980 nm, $1.0$ W cm$^{-2}$, 5 min), and 6 h later, toxicity was measured by using the CCK-8 and Cytotoxicity LDH Assay Kit (Dojindo, following the manufacturer's instructions). The Transwell polycarbonate membrane cell culture insert set (Labselect, $3.0$ µm pore size) was fitted into a 24-well cell culture plate for this study.

**Western blotting**
Western blotting was used to detect the protein expression in tumour cells treated with TIOs and NIR light (980 nm, $1.0$ W cm$^{-2}$, 5 min). TIOs-bound tumour cells were seeded into 6-well plates for 6 h and treated with NIR light (980 nm, $1.0$ W cm$^{-2}$, 5 min). The cells were lysed to extract protein by adding 100 µl of cell lysis buffer per 1 million cells. Briefly, separated proteins were transferred to PVDF membranes (Merck Millipore) by SDS–PAGE. After 10 min of blocking in Protein Free Rapid Blocking Buffer (Epizyme Biotech), the membranes were incubated overnight at 4 °C with diluted primary antibody (1:1000). After washing 5 times with TBST, the membrane was further incubated with diluted anti-rabbit IgG HRP-linked antibody (1:5000) for 90 min. After 5 times washing in TBST, the membrane was immersed in Thermo Scientific SuperSignal West Pico PLUS chemiluminescent substrate and exposed using an Amersham ImageQuant 800 (Cytiva).

For 4T1 cells, the expression of GPX4 (ab125066), was detected. For EMT6 and CT26 cells, the cleavage of GSDME (ab215191) or GSDMD (ab209845 or ab219800) were detected. For testing the VCAM-1 (CST39036) expression in BMDC, BMDM, TC-1, AML12, TCMK-1, 4T1, CT26 and EMT6, the above protocol can be used.

### Tumour targeting, penetration and biodistribution

**MPI imaging.** Tumour cells (4T1, EMT6, CT26, $1 \times 106$ cells) in 25 μl PBS were implanted into the right flank of BALB/c female mice (6-weeks-old). When the tumour volumes reached ~150 mm³ (7–10 days), the mice were randomly divided into five groups ($n = 3$ in every group), then TIOs-Fe₃O₄ ($2 \times 10^6$ units/mouse) was injected intravenously. MPI imaging is performed using an MPI scanner (Magnetic Insight Inc, MOMENTUM). The frequency of MPI is 45 kilohertz. The magnetic gradient intensity of MPI is 0–5.5 T/m. During the experiment, mice were anesthetized with 2% isoflurane at 2 L/min oxygen flow. In addition, tumours and major organs (heart, liver, spleen, lungs, and kidneys) were collected at 6, 24, 48, 72, and 96 h after injection and imaged by an MPI scanner (Magnetic Insight Inc, MOMENTUM). The analysis of MPI imaging data processed by VivoQuant software 2.0.

**Tumour penetration.** To construct the tumour model, $1 \times 10^6$ 4T1 cells in 25 μl of PBS were implanted into the right flank of BALB/c female mice (6-8 weeks). When the tumour volume reached 100 mm³, the tumour-bearing BALB/c female mice were intravenously administered DiI labelled $TIO_S$ ($5 \times 106$ units/mouse). 24 h post injection, tumours were harvested and washed with PBS, followed by a well-established frozen section protocol (Leica CM3050S). The frozen sections of the tumours were analysed and imaged with confocal microscopy (Olympus FV3000). The blood vessels in the tumour were stained with a primary anti-CD31 antibody (1:100, BioLegend, 102402) followed by an Alexa Fluor 488-conjugated secondary antibody (1:400, CST, 4416).

### Light controlled peroxidation in tumours

**To quantify the lipid peroxidation in samples from animals that were intravenously injected with TIOs or the indicated control on day 7.** Each of the treatments was followed by NIR light (980 nm, 1.0 W cm⁻², 5 min) on days 8, 9, and 10. Tumours were collected on day 11, and a single-cell suspension was first prepared as described in the previous section. The suspension was centrifuged, and the cell pellets were washed twice with cell staining buffer, blocked with 0.25 μg of TruStain FcX™ PLUS antibody per 10⁶ cells in a 100 μl volume for 5-10 min at 4 °C, washed and stained with anti-CD45 antibody followed by BODIPY 581/591 C11. Cells were strained through a 40 μM cell strainer and analysed immediately with a flow cytometer. For BODIPY 581/591 C11 staining, the signals from both nonoxidized C11 (PE channel) and oxidized C11 (FITC channel) were monitored. The ratio of the mean fluorescence intensity (MFI) of FITC to the MFI of PE was calculated for each sample. The data were normalized to control samples and shown as the relative lipid ROS. In addition, tumours were collected on day 11 and washed with PBS, followed by a well-established frozen sectioning protocol (Leica CM3050S). To perform lipid peroxidation staining, frozen sections of the tumours were treated with 100 μl DMEM containing 10 μM BODIPY 581/591 C11 and incubated for 30 min at 37 °C in a 5% $CO_2$ incubator. The frozen sections were washed three times with 2 mL PBS and then analysed immediately on an Olympus confocal microscope. The imaging data shown are representative of at least three randomly selected fields.

### In vivo antitumour study

All mice used were purchased from Charles River Laboratories. To construct the tumour model, $1 \times 10^6$ 4T1, EMT6 or CT26 cells in 25 μl of PBS were implanted into the right flank of Nu/Nu nude female mice (for 4T1 cells) or BALB/c female mice (6-8 weeks). Mice were intravenously injected with TIOs ($5 \times 10^6$ units/mouse) or an indicated control on day

7, and each of the treatments was followed by NIR light (980 nm, 0.5 W cm⁻², 5 min) on days 8, 9, and 10. After monitoring tumour growth, the mice were euthanized once the tumours reached ~1000 mm³ in size.

**Evaluation of the immune memory effect of TIOs.** Subcutaneous 4T1 tumour-bearing BALB/c female mice were intravenously injected with TIOs ($5 \times 10^6$ units/mouse) or the indicated control on day 7. After three cycles of treatment with a 980 nm laser (0.5 W cm⁻², 5 min), the mice were inoculated subcutaneously again on the left flank with 4T1 cells ($5 \times 10^5$ cells/mouse). After monitoring tumour growth, the mice were euthanized once the tumours on the right flank reached ~1000 mm³ in size.

**Evaluation of the efficacy of TIOs against tumours in nude mice.** 4T1 cells ($1 \times 10^6$ cells) in 25 μl PBS were implanted subcutaneously into the right flanks of female *nude* mice (6-8 weeks). Nude mice were injected intravenously with TIOs ($5 \times 10^6$ units/mouse) or the indicated controls on day 7, and each of the treatments was followed by NIR light (980 nm, 0.5 W cm⁻², 5 min) on days 8, 9, and 10.

**Evaluation of the efficacy of TIOs+anti-PD-1 in the treatment of tumours.** A 4T1 tumour model was established in female mice (6–8 weeks) using the same method. Mice were injected intravenously with TIOs ($5 \times 10^6$ units/mouse) or the indicated controls on day 7 after tumour inoculation, and the tumours were treated with a 980 nm laser (0.5 W cm⁻², 5 min) on day 8; each treatment was followed by 4 intraperitoneal injections of anti-PD-1 (100 μg/mouse, on days 9, 11, 13 and 15).

### Side-effect analysis in a mouse tumour model

**For histological evaluation.** Tumour-bearing mice were killed after indicated treatments, and kidneys, hearts, lungs, spleens, and livers were removed and fixed with 4% formalin overnight. After dehydration by graded ethanol treatment, tissue samples were embedded in paraffin and sectioned for hematoxylin and eosin staining. Independent experiments were performed with three mice in each experimental group. Data were analyzed by a professional clinician.

**Systemic inflammatory response.** On day 1, 2, 3, 4 and 5 after intravenously injection of TIOs, blood was gained from the eyeballs of mice to detect RBC, WBC and PLT, and serum IL-6 concentration was detected by ELISA. In addition, the rectal temperature and body weight of mice were measured.

### Immune cell depletion and cytokine neutralization

Each tumour-bearing mouse was intraperitoneally administered 200 μg of anti-mouse CD4, anti-mouse CD8 or the isotype-control antibody on days 7, 8, 9 and 10 after inoculation of 4T1 cells. To verify the depletion efficiency, the percentage of CD4⁺ or CD8⁺ T lymphocytes in the spleen was determined on day 16 by using an Agilent Novocyte flow cytometer. For cytokine neutralization experiments, 100 μg of anti-IL-1β (BE0246, BioXcell), anti-IL-18 (BE0237, BioXcell) or anti-HMGB1 (SQab20175, Arigo Biolaboratories) was intraperitoneally injected into each tumour-bearing mouse 3 times (once every 3 days).

### FACS analyses of tumour-infiltrating lymphocytes

To construct the tumour model, 4T1 cells ($1 \times 10^6$ cells/mouse) in 25 μl PBS were implanted into the right flanks of BALB/c female mice (6–8 weeks). Mice were intravenously injected with TIOs or the indicated control on day 7, and each of the treatments was followed by NIR light (980 nm, 1.0 W cm⁻², 5 min) on days 8, 9, and 10. For FACS of tumour-infiltrating lymphocytes, tumours were collected on day 13. The tumours were dissected from the surrounding fascia, weighed, and minced into pieces with a gentleMACS Dissociator (Miltenyi

Biotec). Cell clumps were removed with a 100-µm cell strainer to obtain single-cell suspensions. The suspension was centrifuged, and the cell pellets were washed twice with cell staining buffer (Biolegend), blocked with 0.25 µg of TruStain FcX™ PLUS (anti-mouse CD16/32, BioLegend) antibody per $10^6$ cells in a 100 µl volume for 5-10 min at 4 °C, and incubated with anti-CD45-FITC (S18009F, BioLegend), anti-CD3-APC/Cyanine7 (17A2, BioLegend), anti-CD4-APC (RM4-4, BioLegend) and anti-CD8-PE/Cyanine7 (53−6.7, BioLegend) antibodies at predetermined optimum concentrations (0.25 µg per $10^6$ cells in a 100 µl volume) at 4 °C for 20 min in the dark. The LIVE/DEAD™ Fixable Violet Dead Cell Stain Kit (L34963, Invitrogen) was used to determine cell viability during FACS analysis. The FOXP3 Fixation/Permeabilization Kit (00-5521-00, Invitrogen) was used to stain intracellular FOXP3 following the manufacturer's instructions.

### Cytokine measurements

In vitro, to detect the release of inflammatory factors from the TIOs, Raw 264.7, LNT Raw 264.7, precursors of TIOs and TIOs were subjected to the indicated ELISA treatment at different time points. Raw 264.7 was seeded in a 6-well plate ($10^6$ cells/well) for 12 h, then RSL3@azo-Ce6-mSiO$_2$-UCNPs (100 µl, 300 µg ml$^{-1}$) were added and incubated for 4 h, washed three times with PBS, then 1 ml of fresh medium was added. Suspensions in well plates were collected at different time points (6, 12, 24 and 48 h) and centrifuged at 3000 rpm for 30 min at 4 °C for ELISA. Meanwhile, the prepared LNT Raw 264.7 (liquid nitrogen treated Raw 264.7) and TIOs were thawed from liquid nitrogen and washed three times with PBS. Resuspend the samples in 1 ml of fresh medium and add to a 6-well plate ($10^6$ units/well). The suspension in the well plates were collected at different time points (6, 12, 24 and 48 h) and then centrifuged at 3000 rpm for 30 min with the corresponding ELISA kits (MEIMIAN Biology) to measure IL-1β, IL-6, IL-10, IL-12, IL-18, TNF-α and HMGB1. In addition, the suspension of precursors of TIOs for 6 h and TIOs treated with NIR 980 nm (0.5 Wcm$^{-2}$, 5 min) were collected and centrifuged at 3000 rpm for 30 min at 4 °C for ELISA.

Moreover, tumour homogenates were collected, and orbital blood of mice was collected for coagulation and centrifugation to collect serum. ELISA measurements of IL-1β, IL-18 and HMGB1 concentrations in the serum and tumour homogenates of mice subjected to the indicated treatments were obtained (following the manufacturer's instructions).

### FACS analyses of CD8⁺ T cell stemness

To construct the tumour model, 4T1 cells ($1 \times 10^6$ cells/mouse) in 25 µl PBS were implanted into the right flanks of BALB/c female mice (6−8 weeks). Mice were intravenously injected with TIOs or the indicated control on day 7, and each of the treatments was followed by NIR light (980 nm, 0.5 W cm$^{-2}$, 5 min) on days 8. Subsequently, each tumour-bearing mouse was Intratumorally administered 2.5 mg/kg of cGAMP on day 9, 11, and 13. For FACS of tumour-infiltrating lymphocytes, tumours were collected on day 14. The single-cell suspension was first prepared and blocked as described in the previous section, and incubated with anti-CD45-FITC (S18009F, BioLegend), anti-CD3-APC/Cyanine7 (17A2, BioLegend), anti-CD4-BV786 (GK1.5, BD Biosciences) and anti-CD8-PE/Cyanine7 (53−6.7, BioLegend), antibodies against CD44-PerCP-Cy5.5 (IM7, BD Biosciences), CD62L-BV510 (MEL-14, BD Biosciences), and PD-1-BV605 (J43, BioLegend) antibodies at predetermined optimum concentrations (0.25 µg per $10^6$ cells in a 100 µl volume) at 4 °C for 20 min in the dark. The LIVE/DEAD™ Fixable Violet Dead Cell Stain Kit (L34963, Invitrogen) was used to determine cell viability during FACS analysis (following the manufacturer's instructions). The FOXP3 Fixation/Permeabilization Kit (00-5521-00, Invitrogen) was used to stain intracellular anti-FOXP3-PE (FJK-16s, Invitrogen) and anti-TCF-1-Alexa647 (S33-966, BD Biosciences) following the manufacturer's instructions.

### Single-cell RNA sequencing

FACS analysis of tumour-infiltrating lymphocytes and tumour-infiltrating lymphocytes isolated in tumour models was performed after staining with a PE-conjugated anti-mouse CD45 antibody (clone 30-F11, Biolegend). CD45⁺ immune cells were then enriched using a BD FACS Aria III flow cytometer. Cell viability was monitored in real time during the preparation of single CD45⁺ immune cell suspensions. Ten thousand cells (~600 single cells per microlitre) from each experimental group were barcoded and pooled using a 10x Genomics device. Samples were prepared according to the manufacturer's protocol and sequenced on an Illumina NextSeq sequencer. The Cell Ranger Analysis Pipeline (v.3.0.2) was used for sample demultiplexing, barcode processing, alignment, filtering, UMI counting, and aggregation of sequencing runs. For quality control of the single-cell RNA-sequencing process, cells with <300 genes detected and cells with mitochondrial-encoding gene transcript counts exceeding 15% of total transcript counts were removed from subsequent analyses. Genes detected in less than three cells in the entire dataset were also excluded, resulting in a preliminary expression matrix of 15603 cells. After obtaining the digital gene expression data matrix, dimensionality reduction, clustering and differential gene expression analysis were performed using Seurat (v.3.0.0.9000).

### Statistical analysis

All the values in the present study are presented as the mean ± SD unless otherwise indicated in the figure captions. One-way analysis of variance was used for multiple comparisons when more than two groups were compared, and Student's $t$-test was used for two-group comparisons. All statistical analyses were conducted with the Graph-Pad Prism software package (PRISM 8.0; GraphPad Prism Software), Excel 2019. Survival curves were obtained using the Kaplan−Meier method and compared by the log-rank test. The threshold for statistical significance was $P < 0.05$.

### Animal ethics and general protocols for animal studies

All mouse studies were conducted in accordance with the principles and procedures outlined in the Guide for the Care and Use of Laboratory Animals (Ministry of Health, China), and the protocol was approved by the Animal Experimental Ethical Inspection committee of the Laboratory Animal Centre, Wenzhou Medical University (ID number: xmsq2021-0530). The tumour-bearing mice were subjected to the indicated treatments when the tumour volume reached 100 mm$^3$ (~1 week after inoculation) or 200-300 mm$^3$ (2-3 weeks after inoculation). Ethical compliance with the IACUC protocol was maintained. In none of the experiments did the size of the tumour graft surpass 2 cm in any two dimensions, and no mouse had severe abdominal distension (≥10% increase in original body weight), as outlined by the Animal Experimental Ethical Inspection committee of the Laboratory Animal Centre, Wenzhou Medical University.

### Reporting summary

Further information on research design is available in the Nature Portfolio Reporting Summary linked to this article.

## Data availability

The mass spectrometry proteomics data have been deposited to the ProteomeXchange Consortium with the dataset identifier PXD044912. Single-cell RNA sequencing of tumour-infiltrating immune cells were also deposited at the Gene Expression Omnibus (GEO) under accession number GSE227675. The remaining data supporting the findings of this study are included in the Article, Supplementary Information and Source Data. Source Data are available with the paper. Source data are provided with this paper.

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

## Acknowledgements

This work was supported by National Natural Science Foundation of China grant 22007074 to Q.W. We are grateful for the support of the Science and Technology Innovation Young Top Talents Programme of Wenzhou city. We thank Wenzhou Medical University and the Second Affiliated Hospital of Wenzhou Medical University for start-up funding. We thank the scientific research centre of Wenzhou Medical University for kindly providing technical support.

## Author contributions

Q.W. conceived the study; H.Y. assisted by Y.Z. performed material synthesis, characterization and chemical analysis; Z.Z., H.Y. assisted by C.X. performed imaging and data analysis; Z.Z. performed the pathological analysis; H.Y. performed all other experiments. Z.Z. provided technical assistance (FACS) and valuable suggestions. Q.W., H.Y., Z.Z. revised the study. H.Y., Z.Z and Q.W. analysed the data. Q.W. assisted by Z.Z. wrote the manuscript with input from all authors. All authors discussed the results and commented on the manuscript.

## Competing interests

The authors declare no competing interests.
