## [Peer Review File · Nature Communications]

A Cytotoxic T Cell Inspired Oncolytic Nanosystem Promotes Lytic Cell Death by Lipid Peroxidation And Elicits Antitumor Immune ResponseREVIEWER COMMENTS

Reviewer #1 (Remarks to the Author): with expertise in nanomedicine, cancer, immunology

Zuo et al. described a T-cell-inspired oncolytic system (TIOs) for anticancer therapy by initiating lytic cell death in a NIR light-controlled and catalytic manner in this publication. The authors demonstrated that TIOs are highly effective at tumor targeting and penetration. Following the use of TIOs in anticancer treatments, some tumor models could be eliminated effectively with minimal harm to organs. Overall, this study was well-designed and performed. I support acceptance of this work after addressing the following issues.

1. The enrichment effect of TIO₂ in tumors is impressive but unconventional. The data shown in figure 5a demonstrated that more than 90% of the TIOs accumulated in the tumor after intravenous injection. This result may be unreliable because the means of Dil labeling used could lead to non-specific enrichment. It is recommended that the authors use more precise quantitative means to analyze the biodistribution of TIOs, such as radioelements labeling.
2. I am curious if this system of TIOs is tumor cell-specific. In addition to tumor cells, does it cause other cells in the tumor, such as T cells and macrophages, to form pores on their surfaces, leading to the release of intracellular lysates and, thus, the risk of inflammatory factor storms?
3. The composition of the TIOs system is complicated, including RSL3, azo, Ce6, UCNP, etc. Therefore, how to further improve quality control in the preparation process needs further discussion. In addition, the biosafety of TIOs requires further validation.
4. Several of the elements in the figure, such as the markers for significant difference analysis, might be enhanced further. The term 'transwell' in figure 4g appears superfluous.
5. Schematics need to be further improved for readability and aesthetics.

Reviewer #2 (Remarks to the Author): with expertise in nanomedicine, cancer, immunology

In this manuscript, Prof. Wang established an easily prepared near infrared (NIR) light-controllable micron-scale T-cell-inspired oncolytic system (TIOs) featuring a catalytic, controllable and direct membranolytic ability to manipulate the oncolysis, inflammation, and antitumour immune function. The authors showed a T-cell-inspired oncolytic system efficiently triggered lytic cell death in a NIR light controllable and catalytic manner, which provided new insight for designing artificial systems to break through the natural restrictions to manipulate oncolysis. This work is of great significance. I recommended to accept after major revision.

1. Suggesting to further characterize the morphology of TIOs system by TEM.
2. The heating curve in vitro should be provided by the authors to verify the ROS produced by PDT.
3. The stability of TIOs should be evaluated under different conditions, such as room temperature or 4 °C, not only physiological conditions.
4. The author should specify the tumor volume of the mouse tumor at the beginning of the in vivo experiment.
5. It is incomprehensible the TIOs showed high enrichment in tumor bed at 24, 48 and 72 h, while the temperature presented negligible changes during NIR irradiation, the author should give reasonable explanation.

Reviewer #3 (Remarks to the Author): with expertise in cancer immunotherapy

This manuscript by Zuo et al. aims at developing and characterizing a micro-scale cytotoxicity T cell-inspired system (TIO) for cancer immunotherapy. The authors proposed that chemically modified- macrophages (RAW and BMDMs) can “act” in a similar fashion to cytotoxic T cells. In this study, they show data that aligns with this hypothesis. Overall, the platform and concept are interesting and have the potential for further advanced macrophage-based immunotherapy. However, this reviewer feels that the study could be strengthened by conducting some key experiments to support the conclusions and properly proofreading the manuscript.

- General comments related to the experimental approaches and results:

Overall, the data presented support the hypothesis, but on multiple occasions, proper experimental controls still need to be included. Similarly, some conclusions are not fully supported by the data. Some examples are listed below:

1. The data in figure 2f shows cytokine profiles for the different conditions investigated. The authors directly linked this data to ROS-related stress responses. However, based on the data presented herein, a direct link cannot be made.

2. Data in Fig 2 shows that TIOs can bind to cancer cells, but it would be important to test if TIOs bind (or not) to healthy cells. This is important as in the in vivo biodistribution shown in Fig 5, TIOs can also target normal tissue (albeit at lower concentrations than tumours)

3. Supplementary Fig 1i shows that chemically modified BMDMs and RAW significantly drop cell viability after 8 and 10 hs. Did the authors test later time points? For how long are these cells viable?

4. For example, in Figures 3d-f and 4, controls including tumour cells only (no bound to TIOs) need to be included- We do not know the basal impact of NIR light irradiation on tumour cells.

5. For Supplementary Fig 4 and line 269, where is the data showing that 4T1, EMT6 and CT26 cells are PD-1 positive? While PD-1 can be expressed in some sub-populations of cancer cells, PD1 expression is typically present in various immune cells.

6. It is unclear to this reviewer how the data in Fig 4a clearly shows a "lytic phenotype". Should a positive control be included, or a different and more quantitative assay be considered?

7. Based on the data discussed and displayed in line 292 and Fig 4e, the downregulation of

GPX4 in 4T1+TIOs cells upon treatment with NIR is not apparent. It would be essential to show the densitometry of 3 independent experiments.

8. The authors concluded that Fig 5 (Lines 376-377) shows that TIOs displayed better killing than CD8T cells...but the data does not support this conclusion. when did the authors directly compare TIOs and CD8 T cell killing head-to-head?

9. Based on data in Fig 6, the authors conclude that both CD4+ and CD8+ T cells are “activated” and important for TIOs therapeutic effects.... but the data in Fig 6c is intriguing as depletion of each population of T cells individually is sufficient to abrogate the effects of TIOs completely (and not partially as expected).

10. In Fig 6g-h, an anti-PD1 monotherapy control is missing

- General comments regarding manuscript structure and organization:

1. It was hard to follow the description of the results as the figures on multiple occasions are not described in sequential order (e.g. Fig. 4i is introduced before Fig. 4c)

2. Some panels are missing statistical analysis.

3. Multiple sentences are missing words (e.g., Line 66), and in some cases, the description of the results can be improved to make it easier for the readers to follow.

Reviewer #4 (Remarks to the Author): with expertise in lipid peroxidation

A novel cytotoxic system TIOs was introduced to induce cancer lipid peroxidation or cell death and antitumor immune in mice. Although RSL3-induced lipid peroxidation has been extensively studied, the material carrying RSL3 that specifically recognize tumors is innovative. However, the mechanism how TIOs recognizes tumor cells but not normal cells

need to be explored in more detail. In addition, how TIOs induces antitumor immune should be elucidated. More specifically, there are some concerns that need to be addressed.

Major:

(1) As shown in fig1. I am confused about the term “T-cell-inspired oncolytic system (TIOs)”.

First, T cells recognize tumors, but the recognition of TIOs depends on macrophage, but not T cells. There is no evidence that TIOs bind tumors in a similar way to that of T cells.

Second, T cells induce pores in tumors, but whether TIOs induce pore in tumors is unclear.

The released contents are different. In addition, T cells contents are actively released, but TIOs contents is passively released by NIR. Thus, why not the term is “macrophage-directed oncolytic system”.

(2) Line 380: the author stated that “However, there was no tumour shrinkage when we repeated the above treatments in immunodeficient mice”. Why did RSL3 released by TIOs not induce cell death in tumors in immunodeficient mice, since TIOs induce obvious cell death in vitro. As the authors stated in the title TIOs induce “Lytic Cell Death”, but why not in immunodeficient mice. This phenomenon may just due to that TIOs or RSL3 lose effect in this mouse strain, and may not be related to immunity. At least the authors should test whether TIOs induce lipid peroxidation in Fig.6b. In addition, it is important to test whether supplement T cells decrease tumors in the immunodeficient mice?

(3) In theory, TIOs kill tumors in vitro, it may also directly kill tumor cells in vivo. Thus, it is unknown why authors focus on antitumor immune while tumors can be directly eliminated by TIOs. In addition, it is unclear how TIO induces antitumor immune. Whether TIO induces immunogenic cell death (ICD)? If yes, the release of DAMP molecules (such as HMGB1 and ATP) needs to be examined in vitro and in vivo.

(4) Following the previous question, whether vaccination with TIOs-pretreated cancer cells induces antitumor immunity in mice? If yes, the author should also rechallenge the mice with cancer cells to check the antitumor immunity induced by vaccination.

(5) Fig3b: The authors should test whether TIOs could bind VCAM-1 knockout tumour cells. Negative control (such as normal breast cells) should also be included. Otherwise, it's just the TIOs and tumor cells getting close to each other, not the real “bind”. In addition, the cell death inducing activity of TIOs in VCAM-1 knockout tumour cells and normal breast cells should be tested.

Minor:

(1) Title: "A Cytotoxic T Cell Inspired Oncolytic System for Manipulating Lytic Cell Death Induced Antitumor Immune Function by Direct Lipids Peroxidation". It is difficult to distinguish between "direct" and "indirect" induced by RSL3. Thus, "direct" may be deleted. Furthermore, "Lipids Peroxidation" may be replaced by the more common term "lipid peroxidation".

(2) Abstract: The author stated that TIO did not induce inflammatory: "The TIOs exhibited exceptional tumour targeting and penetration without any inflammatory risk" (line 42), paradoxically, the author also stated that TIO induced inflammatory: "Tumour regression was correlated with oncolysis-mediated inflammation" (line 45).

(3) Line 255 "the ROS was accumulating in the plasm membrane". This description is not accurate because there is no marker of plasm membrane and it seems that ROS is widely distributed in the cytoplasm as shown in Fig3D.

(4) Line 295: "RLS3" should be "RSL3".

(5) Fig1b: RSL3 should be included in the figure1. But whether TIOs induce pores in tumor is unknown.

(6) Fig2a: "proes" should be "pores".

(7) A clearer distinction is needed in Fig6b and Fig6c legends.

Point to Point Response

- Reviewer #1 (Remarks to the Author): with expertise in nanomedicine, cancer, immunology

Zuo *et al.* described a T-cell-inspired oncolytic system (TIOs) for anticancer therapy by initiating lytic cell death in a NIR light-controlled and catalytic manner in this publication. The authors demonstrated that TIOs are highly effective at tumor targeting and penetration. Following the use of TIOs in anticancer treatments, some tumor models could be eliminated effectively with minimal harm to organs. Overall, this study was well-designed and performed. I support acceptance of this work after addressing the following issues.

We appreciated your evaluation; your constructive suggestions significantly improved our study. Next, we will answer your concerns point-to-point.

1. The enrichment effect of TIOs in tumors is impressive but unconventional. The data shown in figure 5a demonstrated that more than 90% of the TIOs accumulated in the tumor after intravenous injection. This result may be unreliable because the means of Dil labeling used could lead to non-specific enrichment. It is recommended that the authors use more precise quantitative means to analyze the biodistribution of TIOs, such as radioelements labeling.

Thanks for your suggestions. We found that the problem you pointed out was correct, and Dil underwent non-specific enrichment. Therefore, we sincerely apologized for the misunderstanding caused by our unprofessional labeling method. (See the line. 336-368 in our revised manuscript. Red font.)

- To correct this major mistake, we used magnetic particle imaging (MPI, 10.1021/acsnano.9b08660) to image TIOs. Fe_3O_4 nanoparticles and RSL3@azo-Ce6-mSiO₂-UCNPs were swallowed together into Raw 264.7 cells to get TIOs- Fe_3O_4 (Revision Data Fig. 1a). Then, the TIOs- Fe_3O_4 were intravenously injected into the 4T1, EMT6 and CT26 tumour bearing mice.
- To our delight, the results showed that TIOs were still effective at tumor targeting (Revision Data Fig. 1b,c,g,h and 2a-b). In absolute terms, the enrichment of TIOs reached peaks at 24 hours post-injection, and then gradually decreased. The tumor to liver ratio reached peaks at 72 hours (the peak value was about 4.5) (Revision Data Fig. 1d,e,i,j and 2c,d,f). Meanwhile, to exclude non-specific enrichment of MPI imaging agents themselves, we also injected imaging agents alone into tumor-bearing mice and imaged them. The results showed that in the absence of TIOs, the imaging agents themselves mainly enriched in the liver (Revision Data Fig. 1f,k and 2e). Thus, the results of MPI imaging supported the selective tumor-targeting of TIOs.

2. I am curious if this system of TIOs is tumor cell-specific. In addition to tumor cells, does it cause other cells in the tumor, such as T cells and macrophages, to form pores on their surfaces, leading to the release of intracellular lysates and, thus, the risk of inflammatory

factor storms?

Thanks for your questions and suggestions. (See the line. 227-228 and line. 399-404 in our revised manuscript. Red font.)

- For testing if Dil labeled TIOs could bind immune cells, we collected and cultured the BMDCs (primary dendritic cells) and iBMDMs (primary bone marrow macrophages) from BALB/c mice, then, we added TIOs to BMDCs and BMDMs *in vitro*, respectively (Revision Data Fig. 2g). After incubation and PBS washing, the images showed TIOs failed to bind the BMDCs or BMDMs. Further, we tested if Dil labeled TIOs could bind healthy cells (e.g., tissue culture-1, alpha mouse liver 12, transformed C3H mouse kidney-1), after PBS washing, we took photos. The results showed that TIOs failed to bind to the indicated healthy cells, which may be due to the low express of VCAM-1 of these cells (Revision Data Fig. 2g-h).
- For *in vivo* situation, as you mentioned, in tumors, its difficult to image if TIOs only recognize tumor cells *in vivo*. However, we tested the risk of inflammatory factor storms by detecting the body weight, IL-6 release and body temperature after TIOs-mediated oncolysis. To our delight, no fever, IL-6 release, and body weight loss were found for 3 consecutive days after treatments (Revision Data Fig. 2i-l), indicating little risk of inflammatory factor storms.

3. The composition of the TIOs system is complicated, including RSL3, azo, Ce6, UCNP, etc. Therefore, how to further improve quality control in the preparation process needs further discussion. In addition, the biosafety of TIOs requires further validation.

Thanks for your suggestions. (See the line. 166, 192-197 in our revised manuscript. Red font.)

- We further characterized the loading capacity of RSL3, azo, Ce6, and at the same time, we performed TEM imaging on TIOs, which showed significant RSL3@azo-Ce6-mSiO₂-UCNPs enrichment inside the TIOs (Revision Data Fig. 3a).
- Meanwhile, for safety, we injected TIOs into BALB/c mice and monitored their blood terms, IL-6, and body temperature for 5 consecutive days. The results showed that compared with the PBS group, the TIOs treatment group showed no significant changes in the above indicators, indicating that our TIOs themselves were safe and non-proinflammatory (Revision Data Fig. 3b-h).

4. Several of the elements in the figure, such as the markers for significant difference analysis, might be enhanced further. The term transwell in figure 4g appears superfluous.

Thanks for your questions and suggestions.

- We have reviewed the manuscript and enhanced the significant difference analysis (e.g., Fig. 2e,g, 4i, Supplementary Fig. 4d, 5f, 18c). As you suggested, we delete the term "transwell" in figure 4g (revised version Fig. 4h).

5. Schematics need to be further improved for readability and aesthetics.

Thanks for your questions and suggestions.

- We have updated the schematics (Revision Data Fig. 3i).
- Reviewer #2 (Remarks to the Author): with expertise in nanomedicine, cancer, immunology

In this manuscript, Prof. Wang established an easily prepared near infrared (NIR) light-controllable micron-scale T-cell-inspired oncolytic system (TIOs) featuring a catalytic, controllable and direct membranolytic ability to manipulate the oncolysis, inflammation, and antitumour immune function. The authors showed a T-cell-inspired oncolytic system efficiently triggered lytic cell death in a NIR light controllable and catalytic manner, which provided new insight for designing artificial systems to break through the natural restrictions to manipulate oncolysis. This work is of great significance. I recommended to accept after major revision.

We appreciated your evaluation; your constructive suggestions significantly improved our study. Next, we will answer your concerns point-to-point.

1. Suggesting to further characterize the morphology of TIOs system by TEM.

Thanks for your suggestions. (See the line. 184-187 in our revised manuscript. Red font.)

- We performed TEM imaging for TIOs, and the results showed RSL3@azo-Ce6-mSiO₂-UCNPs were clustered in the interior of TIOs. Moreover, the nuclei of TIOs were still visible, and damages in the cell membrane caused by freezing could be seen (Revision Data Fig. 4a. The white box shows cell membrane damage).

2. The heating curve *in vitro* should be provided by the authors to verify the ROS produced by PDT.

Thanks for your suggestions. (See the line. 293-295 in our manuscript. Red font.)

- We followed your suggestion to irradiate PBS, TIOs, and RSL3@azo-Ce6-mSiO₂-UCNPs with 980 nm NIR light and recorded their temperature change to investigate the heating curve. The results showed that as the irradiation time or power increased, the temperature slowly increased (Revision Data Fig. 4b).
- Notably, for *in vitro* assays in our study, the NIR light we employed was only 1.0 Wcm⁻² (5 min). At this condition, the photothermal effect of PBS, TIOs, and RSL3@azo-Ce6-mSiO₂-UCNPs was not significant, almost only 0.5 °C change (Revision Data Fig. 4b).
- Meanwhile, in our manuscript, Fig. 2h,i and Supplementary Fig. 1h recorded the relationship between ROS generation and irradiation time (Revision Data Fig. 4c).

3. The stability of TIOs should be evaluated under different conditions, such as room temperature or 4 °C, not only physiological conditions.

Thanks for your suggestions.

We followed your suggestion to evaluate the stability of TIOs under room temperature and 4 °C. The results showed that TIOs were also stable under the above conditions (Revision Data Fig. 4d or Supplementary Fig. 1l in our revised manuscript).

4. The author should specify the tumor volume of the mouse tumor at the beginning of the *in vivo* experiment.

Thanks for your suggestions.

We labeled the tumor volume of mice at the beginning of the *in vivo* experiments.

5. It is incomprehensible the TIOs showed high enrichment in tumor bed at 24, 48 and 72 h, while the temperature presented negligible changes during NIR irradiation, the author should give reasonable explanation.

Thanks for your suggestions. (See the line. 336-368 and line. 396 in our revised manuscript. Red font.)

- First, we apologized for the misunderstanding caused by the biological distribution of TIOs, because according to the suggestion of reviewer #1, we found that Dil labeling would produce non-specific enrichment. Thus, we used magnetic particle imaging to update the biological distribution of TIOs. As shown in the “Revision Data Figure 1 and 2”, TIOs still enriched in tumors. In absolute terms, the enrichment of TIOs reached peaks at 24 hours post-injection, and then gradually decreased. The tumor to liver ratio reached peaks at 72 hours (the peak value was about 4.5). Meanwhile, to exclude non-specific enrichment of MPI imaging agents themselves, we also injected imaging agents alone into tumor-bearing mice and imaged them. The results showed that in the absence of TIOs, the imaging agents themselves only enriched in the liver.
- Moreover, we imaged the thermal effect of TIOs in tumors under NIR light (Revision Data Fig. 4e). The results showed the thermal effect of TIOs+NIR group and NIR alone group was not obvious under the light intensity of 0.5 Wcm⁻² (the irradiation protocol used in this study); However, when the light intensity increased to 1.0 Wcm⁻², the photothermal effect of 980 nm light would increase significantly. Therefore, in our study, the reason why thermal effect was unobvious might mainly because the NIR light power we used to be low.

➤ Reviewer #3 (Remarks to the Author): with expertise in cancer immunotherapy

This manuscript by Zuo et al. aims at developing and characterizing a micro-scale cytotoxicity T cell-inspired system (TIO) for cancer immunotherapy. The authors proposed that chemically modified- macrophages (RAW and BMDMs) can “act” in a similar fashion to cytotoxic T cells. In this study, they show data that aligns with this hypothesis. Overall, the platform and concept are interesting and have the potential for further advanced macrophage-based immunotherapy. However, this reviewer feels that the study could be

strengthened by conducting some key experiments to support the conclusions and properly proofreading the manuscript.

- General comments related to the experimental approaches and results:

Overall, the data presented support the hypothesis, but on multiple occasions, proper experimental controls still need to be included. Similarly, some conclusions are not fully supported by the data. Some examples are listed below:

We appreciated your evaluation; your constructive suggestions significantly improved our study. Next, we will answer your concerns point-to-point.

1. The data in figure 2f shows cytokine profiles for the different conditions investigated. The authors directly linked this data to ROS-related stress responses. However, based on the data presented herein, a direct link cannot be made.

Thanks for your criticism. (The order has been changed to Fig. 2g in our revised manuscript)

- We apologized for our unclear description; we supplemented an in-depth analysis of this image here.
- In fact, the purpose of this figure is to explain why we need to use liquid nitrogen freezing treatment in the construction of TIOs instead of directly using live macrophages. We believe that this was mainly due to two reasons: firstly, in the living cell system (precursors of TIOs), RSL3@azo-Ce6-mSiO₂-UCNPs themselves interacted with living macrophages, resulting in the release of many inflammatory factors (black columns); Secondly, in our study, RSL3@azo-Ce6-mSiO₂-UCNPs were designed as NIR light controlled nanoparticles, which produced ROS under NIR irradiation (Fig. 2h,i and Supplementary Fig. 1h record the relationship between ROS generation and irradiation time). In the case of with NIR (red columns), precursors of TIOs produced more inflammatory factors than other groups, indicating an obvious ROS-related stress responses in living macrophages. (we added the above explanation in the legend)

2. Data in Fig 2 shows that TIOs can bind to cancer cells, but it would be important to test if TIOs bind (or not) to healthy cells. This is important as in the *in vivo* biodistribution shown in Fig 5, TIOs can also target normal tissue (albeit at lower concentrations than tumours)

Thanks for your suggestion. (See the line. 227-228, 192-197 and 399-404 in our revised manuscript. Red font.)

- We incubated TIOs with healthy cells (e.g., tissue culture-1, alpha mouse liver 12, transformed C3H mouse kidney-1), and after PBS washing, we took photos (Revision Data Fig. 5a). The results showed that TIOs failed to bind to the indicated healthy cells, which may be due to the low express of VCAM-1 of these cells.
- Moreover, we agreed that TIOs also enriched in healthy tissues, for example, livers. However, our NIR light only irradiated tumors, in tissues without light, TIOs could not

be used for oncolysis, thus, for TIOs alone treated mice, no proinflammatory effects were found (Revision Data Fig. 3b-h).

3. Supplementary Fig 1i shows that chemically modified BMDMs and RAW significantly drop cell viability after 8 and 10 hs. Did the authors test later time points? For how long are these cells viable?

Thanks for your suggestion.

- We increased the detection time point, and the results showed that the longer the incubation time, the more obvious the cell death (Revision Data Fig. 5b).
- As mentioned in our manuscripts and Supplementary Fig. 1i, the coincubation of RSL3@azo-Ce6-mSiO₂-UCNPs and macrophages (Raw 264.7 or BMDMs) for four hours could generate ideal UCNP-engineered macrophages by balancing the loading capacity and cell viability. Therefore, after four hours, we ended co-incubation and directly treated the precursors of TIOs with liquid nitrogen to generate the TIOs.

4. For example, in Figures 3d-f and 4, controls including tumour cells only (no bound to TIOs) need to be included- We do not know the basal impact of NIR light irradiation on tumour cells.

Thanks for your suggestion. (*See the line. 263-265, 291-295 in our manuscript. Red font.*)

- We supplemented the effect of NIR light alone on the viability of 4T1, EMT6 and CT26 cells, and the results showed that under our irradiation conditions (1.0 Wcm⁻², 5 min), NIR light itself did not affect cell viability (Revision Data Fig. 5c). Moreover, for lipids peroxidation and cell morphology, NIR light alone (1.0 Wcm⁻², 5 min) would not cause lipids peroxidation (Revision Data Fig. 6a). The above data indicated NIR light alone had little effects on tumour cells.
- Although 980 nm NIR light was considered to cause thermal effect, it depended on the irradiation time and intensity. The irradiation condition used in our study was "5 mins, 1.0 Wcm⁻²" and no obvious thermal effect was detected under this condition (Revision Data Fig. 4b). In addition, many studies of nanomedicine had shown that unless nano-materials (e.g., gold nanorods, graphene, quantum dots, <https://doi.org/10.1021/acs.chemrev.3c00159>) were used to enhance the photothermal effect, in general, NIR light alone would not damage cells.

5. For Supplementary Fig 4 and line 269, where is the data showing that 4T1, EMT6 and CT26 cells are PD-1 positive? While PD-1 can be expressed in some sub-populations of cancer cells, PD1 expression is typically present in various immune cells.

We greatly appreciated your criticism and sincerely apologized for our mistake. (*See the line. 282-283 in our manuscript. Red font.*)

- In fact, we tried to describe that 4T1, CT26, and EMT6 cells are PD-L1 positive. We added FACS analysis (Revision Data Fig. 5d).

6. It is unclear to this reviewer how the data in Fig 4a clearly shows a "lytic phenotype".

Should a positive control be included, or a different and more quantitative assay be considered?

Thanks for your suggestion. (See the line. 287-289 in our revised manuscript. Red font.)

- First, we sincerely apologized for our inappropriate description. We corrected "lytic phenotype" to "morphology of lytic cell death". Secondly, according to our knowledge, pyroptosis and ferroptosis (<https://doi.org/10.1038/s41392-022-01110-y>), two typical lytic cell death, were characterized by cell swelling and bubbling. Our tumour cells treated with TIOs+NIR showed similar morphology (Fig. 4a), thus, we said "morphology of lytic cell death".
- Moreover, in our study, we mainly used LDH leakage as an indicator for quantitative assays, because when the cell membrane ruptured, a large amount of LDH would be released, while, when the cell membrane kept intact, the LDH release would be low (Fig. 4b-e, i). The LDH release was also used in the detection of other lytic cell death (e.g., pyroptosis), thus, we thought it should be an appropriate means to characterize our TIOs-mediated cell lysis (ref. 6-8).

7. Based on the data discussed and displayed in line 292 and Fig 4e, the downregulation of GPX4 in 4T1+TIOs cells upon treatment with NIR is not apparent. It would be essential to show the densitometry of 3 independent experiments.

Thanks for your suggestion. (See the line. 310-311 in our revised manuscript. Red font.)

- We conducted three parallel experiments on GPX4 and quantitatively analyzed the WB results. The results showed that the down regulation of GPX4 in 4T1+TIOs cells on treatment with NIR was statistically significant (Revision Data Fig. 6b,c).

8. The authors concluded that Fig 5 (Lines 376-377) shows that TIOs displayed better killing than CD8T cells...but the data does not support this conclusion. when did the authors directly compare TIOs and CD8 T cell killing head-to-head?

Thank you for your suggestions. (See the line. 407-410 in our revised manuscript. Red font.)

- We apologized for our unclear description. We have strengthened the description. In fact, we tried to describe that because these tumors were PD-L1 positive (Revision Data Fig. 5d), the killing efficiency of natural CD8⁺T cells was very low, therefore, without intervention (e.g., PBS), the tumors grew rapidly. However, TIOs could execute the killing tasks and clear tumours without being interfered by immune checkpoints inhibition, thus, TIOs showed better killing ability than CD8⁺ T cells.
- We conducted a head-to-head analysis *in vitro* to compare the killing ability of TIOs and CD8⁺ T cells on attacking PD-L1 positive tumour cells (Revision Data Fig. 6d). For a clearer observation and statistical analysis of the killing effect, we selected gasdermin E positive tumor cells (EMT6) as the target cells. In principle, both CD8⁺T cells and TIOs+NIR caused pyroptosis of EMT6 cells, leading to the bubbling phenotype. In this experiment, EMT6 cells were treated by CD8⁺T cells and TIOs+NIR,

respectively. Delayed photography showed that pyroptosis mediated by TIOs+NIR was more obvious. Therefore, TIOs showed better killing ability than CD8⁺ T cells on attacking PD-L1 positive tumour cells.

9. Based on data in Fig 6, the authors conclude that both CD4⁺ and CD8⁺ T cells are “activated” and important for TIOs therapeutic effects.... but the data in Fig 6c is intriguing as depletion of each population of T cells individually is sufficient to abrogate the effects of TIOs completely (and not partially as expected).

Thank you for your criticisms and concerns.

- Your concern is reasonable, but we would like to explain it as follows. In this regard, our understanding was that the antitumor immune function induced by TIOs+NIR³ treatment would cause a complex immune cells cooperation. Specifically, our data showed whether CD4⁺ T or CD8⁺ T cells were blocked, the therapeutic effect disappeared completely, indicating that either CD4⁺ T cells or CD8⁺ T cells was important for antitumour immune function, maybe they were in a synergistic state in the TIOs induced antitumor immunotherapy. Moreover, previous study found spontaneous and immunotherapy-induced anti-tumour responses required the activity of both tumour-antigen-specific CD8⁺ and CD4⁺ T cells (<https://doi.org/10.1038/s41586-019-1671-8>). We added the above reference and hope these explanations could reduce your concerns.

10. In Fig 6g-h, an anti-PD1 monotherapy control is missing.

Thank you for your suggestions. (*See the line. 509 in our revised manuscript. Red font.*)

- In fact, Supplement Fig 16c showed an anti-PD1 monotherapy control.

- General comments regarding manuscript structure and organization:

Thank you for your suggestions.

1. It was hard to follow the description of the results as the figures on multiple occasions are not described in sequential order (e.g. Fig. 4i is introduced before Fig. 4c)

Based on your suggestion, we have adjusted the order of most of the figures for easy reading. If there are still parts that need to be improved, we sincerely hope to get your guidance.

2. Some panels are missing statistical analysis.

We supplemented these statistical analyses (e.g., Fig. 2e, g, 4i, Supplementary Fig. 4d, 5f, 18c).

3. Multiple sentences are missing words (e.g., Line 66), and in some cases, the description

of the results can be improved to make it easier for the readers to follow.

Based on your suggestion. We checked the paper and revised the obstacles. However, since our language ability still needs to be improved, if there are still parts that need to be improved, we sincerely hope to get your guidance.

➤ Reviewer #4 (Remarks to the Author): with expertise in lipid peroxidation

A novel cytotoxic system TIOs was introduced to induce cancer lipid peroxidation or cell death and antitumor immune in mice. Although RSL3-induced lipid peroxidation has been extensively studied, the material carrying RSL3 that specifically recognize tumors is innovative. However, the mechanism how TIOs recognizes tumor cells but not normal cells need to be explored in more detail. In addition, how TIOs induces antitumor immune should be elucidated. More specifically, there are some concerns that need to be addressed.

We appreciated your evaluation; your constructive suggestions significantly improved our study. Next, we will answer your concerns point-to-point.

Major:

(1) As shown in fig1. I am confused about the term "T-cell-inspired oncolytic system (TIOs)". First, T cells recognize tumors, but the recognition of TIOs depends on macrophage, but not T cells. There is no evidence that TIOs bind tumors in a similar way to that of T cells. Second, T cells induce pores in tumors, but whether TIOs induce pore in tumors is unclear. The released contents are different. In addition, T cells contents are actively released, but TIOs contents is passively released by NIR. Thus, why not the term is "macrophage-directed oncolytic system".

Thank you for your suggestion and apologized for your confusion.

- We appreciated your point of view. Our TIOs indeed used macrophages as carriers. Compared with the T-cells based tumour recognition and killing, TIOs were different in specific molecules.
- However, we still want to further explain why we use the term "T-cell-inspired oncolytic system (TIOs)" here. First, our original idea of this study was inspired from T cell mediated lytic cell death in tumours (e.g., pyroptosis, ferroptosis, ref. 6, 23). In our study, we tried to build an artificial system to simulate this "killing approach", which was inspired from T-cells, and different from the oncolytic viruses or nanomedicines. Specifically, the attack of T cells on tumor cells was carried out in a catalytic manner outside the cell, and our TIOs also possess this characteristic. Although the carriers of TIOs were macrophages in this study, the carriers could be extended to DCs or T cells (<https://doi.org/10.1038/s41565-021-00972-7>), or even some micron sized functionalized polymers or liposomes.
- Secondly, in T cells mediated killing, after recognition, many signalling pathways in T cells were activated to determine whether to concentrate and transport the cytotoxins to cancer cells (10.1126/science.aay9207). Our NIR light just to mimic the above

approach, that is, when TIOs entered the tumor, NIR light started killing just like "signal pathway activation". However, for non-tumor areas (e.g., livers), due to the absence of NIR light, TIOs would not play a killing function.

- Moreover, we think "inspired" means a source of inspiration (<https://doi.org/10.1038/s44222-022-00010-8>). It was not to claim that our TIOs worked completely the same as T cells, which only learned and drawn lessons from the "mode" of T cells mediating attacking. On specific molecules, such as recognition molecules, toxins, etc., these were variable and scalable for artificial systems. As you suggested, how to make TIOs more like T cells, such as using the same binding molecules and toxins, would be our future goals.
- Finally, although we appreciate the term is "macrophage-directed oncolytic system", we still felt that TIOs was more suitable for explaining the principles and functions of our artificial system. If possible, we hope to continue to use the term "T-cell-inspired oncolytic system (TIOs)".

(2) Line 380: the author stated that "However, there was no tumour shrinkage when we repeated the above treatments in immunodeficient mice". Why did RSL3 released by TIOs not induce cell death in tumors in immunodeficient mice, since TIOs induce obvious cell death *in vitro*. As the authors stated in the title TIOs induce "Lytic Cell Death", but why not in immunodeficient mice. This phenomenon may just due to that TIOs or RSL3 lose effect in this mouse strain, and may not be related to immunity. At least the authors should test whether TIOs induce lipid peroxidation in Fig.6b. In addition, it is important to test whether supplement T cells decrease tumors in the immunodeficient mice?

Thank you for your suggestion. Your question is very instructive. We explain it as follows:

- First, in our study, when TIOs bound tumour cells were irradiated by NIR light, TIOs would produce ROS and release RSL3, then, ROS and RSL3 were transported to the tumour cells and jointly led to cell lysis.
- Secondly, according to your suggestion, we did tumor lipid peroxidation staining in Nu/Nu mice. The results showed that in immunodeficient Nu/Nu mice, TIOs could still cause lipid peroxidation (Revision Data Fig. 7a-c).
- Although *in vitro* experiments, TIOs could very well trigger tumor cell death in dishes, in fact, these results need to consider the relative amount of TIOs and tumor cells. Specifically, *in vitro*, the TIOs in the culture dish were multiples of tumor cells. Therefore, almost all tumor cells were attacked by TIOs after NIR light illumination. However, *in vivo*, TIOs were injected only once (not continuously), thus, the relative amount of TIOs was very low. At this time, the tumor treatment caused by TIOs+NIR³ treatment was more likely due to antitumor immune response.
- We believed that the reason why TIOs failed in mediating tumor treatment in Nu/Nu mice was mainly caused by immune deficiency. Because, compared with *in vitro* experiments, there were few TIOs in tumors, thus, *in vivo*, three times irradiation only led to a small number of lytic cell death. In immune normal mice, a small number of lytic cell death triggered T-cell based anti-tumor immune function (Fig. 6d,e, i, m), so

TIOs cleared the tumors. However, in immune deficient mice, there were no such T cells, which caused the failure of treatment.

- For supplementing T cells (e.g., CAR-T cells), we believed it was a good suggestion. However, we have the following concerns. First, the TIOs+NIR³ treatment triggered local inflammation at the tumor site, thus turning a “cold” tumor into a “hot” tumor by enhancing the T cell infiltration. But, the reinfusion of CAR-T cells might lead to direct tumor clearance, which seemed to be inconsistent with the immune response induced by TIOs+NIR³ treatment. Secondly, our study found that the antitumor immunity induced by TIOs+NIR³ was a complex cooperation of immune cells, such as CD4⁺ and CD⁺ 8 T cells were important for tumor clearance (Fig. 6d). Therefore, it was also worth considering which kind of T cells should be reinfused. Even after T cell transfusion in the immunodeficient mice, it was also unknown whether these T cells could match the inflammation induced by TIOs and ultimately triggered similar antitumor effects.

(3) In theory, TIOs kill tumors *in vitro*, it may also directly kill tumor cells *in vivo*. Thus, it is unknown why authors focus on antitumor immune while tumors can be directly eliminated by TIOs. In addition, it is unclear how TIO induces antitumor immune. Whether TIO induces immunogenic cell death (ICD)? If yes, the release of DAMP molecules (such as HMGB1 and ATP) needs to be examined *in vitro* and *in vivo*.

Thanks for your suggestion.

- As you said, TIOs+NIR could kill tumour cells *in vitro*. However, when it worked *in vivo*, considering that the relative amount of TIOs at the tumor site was relatively low, the reason for tumor clearance might not be the direct killing of all tumour cells, but to activate anti-tumor immunity by inducing lytic cell death-mediated inflammation (line. 455-466).
- We agreed that TIOs+NIR³ might induce immunogenic cell death (line. 455-466). In fact, we have tested many DAMPs *in vivo*, such as IL-1 and HMGB-1 (Supplementary Fig. 17a, b). In addition, as in the scRNA-Seq analyses, the population of monocytes and M1 macrophages increased significantly in TIOs+NIR³ treated 4T1 tumour-bearing BALB/c mice, which indicated a typical inflammatory symptom (Supplementary Fig. 15c). Finally, according to your suggestion, we have supplemented HMGB-1 and ATP testing *in vitro* (Revision Data Fig. 7d).

(4) Following the previous question, whether vaccination with TIOs-pretreated cancer cells induces antitumor immunity in mice? If yes, the author should also rechallenge the mice with cancer cells to check the antitumor immunity induced by vaccination.

Thank you for your suggestion.

- In fact, in our study, we showed a vaccine experiment (line. 447-454, Supplementary Fig. 17). “Encouraged by the TIOs-induced antitumour immune response, we explored immune memory and vaccination effect in BALB/c mice engrafted subcutaneously with 4T1 tumour cells. On day 11, 4T1 cells were subcutaneously

injected again into the left sides of mice that received complete TIOs+NIR³ treatments or PBS treatment (Supplementary Fig. 17). Encouragingly, the mice treated with TIOs+NIR³ had the ability to resist the second transplanted tumours for 50 days, while the tumours grew rapidly on both sides of PBS-treated mice (Supplementary Fig. 17)".

(5) Fig3b: The authors should test whether TIOs could bind VCAM-1 knockout tumour cells. Negative control (such as normal breast cells) should also be included. Otherwise, its just the TIOs and tumor cells getting close to each other, not the real "bind". In addition, the cell death inducing activity of TIOs in VCAM-1 knockout tumour cells and normal breast cells should be tested.

Thank you for your suggestion.

- We used several VCAM-1-negative cells for binding experiments, and the results showed that these cells could not be bound by TIOs (Revision Data Fig. 2g,h).
- In addition, we have made a functional bead (10 uM) on which VCAM-1 proteins were linked, and we tried to employed the above beads to dock the TIOs to explore whether VCAM-1 could serve as a key molecule for bind TIOs. The results showed that the beads with VCAM-1 successfully captured the TIOs, but the beads with albumin failed to capture the TIOs, indicating the recognition between VCAM-1 and integrins could cause the binding (Revision Data Fig. 8).
- Finally, we thought your suggestions were very constructive, although the above results and many previous studies (ref. 31, 32) have shown that the binding of VCAM-1 and integrins was the basis of macrophage and cancer cell recognition, we could not rule out whether other molecules were involved in the binding of TIOs and tumour cells (e.g., PD-1 and PD-L1).

Minor:

(1) Title: "A Cytotoxic T Cell Inspired Oncolytic System for Manipulating Lytic Cell Death Induced Antitumor Immune Function by Direct Lipids Peroxidation". It is difficult to distinguish between "direct" and "indirect" induced by RSL3. Thus, "direct" may be deleted. Furthermore, "Lipids Peroxidation" may be replaced by the more common term "lipid peroxidation".

Thank you for your suggestion. We deleted "direct" and replace the "Lipids Peroxidation" to "Lipid Peroxidation".

(2)Abstract: The author stated that TIO did not induce inflammatory: "The TIOs exhibited exceptional tumour targeting and penetration without any inflammatory risk" (line 42), paradoxically, the author also stated that TIO induced inflammatory: "Tumour regression was correlated with oncolysis-mediated inflammation" (line 45).

Thank you for your suggestion. We sincerely apologize to the misunderstanding caused by our description. In fact, we tried to describe in the case of TIOs themselves without NIR

light, there was no proinflammatory effect.

(3) Line 255 “the ROS was accumulating in the plasm membrane”. This description is not accurate because there is no marker of plasm membrane and it seems that ROS is widely distributed in the cytoplasm as shown in Fig3D.

Thank you for your suggestion. We deleted the “the ROS was accumulating in the plasm membrane”.

(4) Line 295: “RLS3” should be “RSL3”.

Thank you for your suggestion. We replace the RLS3 to RSL3. Since our language ability still needs to be improved, if there are still parts that need to be improved, we sincerely hope to get your guidance.

(5) Fig1b: RSL3 should be included in the figure1. But whether TIOs induce pores in tumor is unknown.

Thank you for your suggestion. We revised the figure (Revision Data Fig. 3i).

(6) Fig2a: “proes” should be “pores”.

Thank you for your suggestion. We revised the figure. Since our language ability still needs to be improved, if there are still parts that need to be improved, we sincerely hope to get your guidance.

(7) A clearer distinction is needed in Fig6b and Fig6c legends.

Thank you for your suggestion. We revised the legends as “**b**, Average tumour volume of TIOs+NIR³ treatment in tumour bearing Nu/Nu mice. **c**, Average tumour volume of TIOs+NIR³ treatment in T cell depletion assays.”

Revision Data Figure 1

Revision Data Figure 2

a

Sample	Loading rate (Wt%)
Ce6	6.73 ± 0.018
azo	2.28 ± 0.021
RSL3	5.32 ± 0.006

i

Revision Data Figure 3

a

TiOs

b

c

d

e

Revision Data Figure 6

a

b

c

d

Point to point response figure legends, materials and methods

Figure legends

Revision Data Figure 1. **a**, Schematic diagram of the structure of Fe_3O_4 labeled TIOs. **b**, The MPI images of three independent 4T1 tumour bearing mice intravenously injected with TIOs- Fe_3O_4 for 96 hours. **c**, The biodistribution of TIOs at different time points in 4T1 tumour bearing mice. H, heart, Lu, lung, S, spleen, L, liver, K, kidney, T, tumour. **d**, Average intensity of TIOs- Fe_3O_4 in the corresponding organs of the indicated tumour models at a series of time points. **e**, The tumour to liver ratio of MPI intensity at different time points. **f**, The MPI images of three independent 4T1 tumour bearing mice intravenously injected with Fe_3O_4 -DHCA for 48 hours. Data are shown as mean \pm s.d. ($n = 3$, independent experiments). **g**, The MPI images of three independent EMT6 tumour bearing mice intravenously injected with TIOs- Fe_3O_4 for 96 hours. **h**, The biodistribution of TIOs at different time points in EMT6 tumour bearing mice. H, heart, Lu, lung, S, spleen, L, liver, K, kidney, T, tumour. **i**, Average intensity of TIOs- Fe_3O_4 in the corresponding organs of the indicated tumour models at a series of time points. **j**, The tumour to liver ratio of MPI intensity at different time points. **k**, The MPI images of three independent EMT6 tumour bearing mice intravenously injected with Fe_3O_4 -DHCA for 48 hours. Data are shown as mean \pm s.d. ($n = 3$, independent experiments).

Revision Data Figure 2. **a**, The MPI images of three independent CT26 tumour bearing mice intravenously injected with TIOs- Fe_3O_4 for 96 hours. **b**, The biodistribution of TIOs at different time points in CT26 tumour bearing mice. H, heart, Lu, lung, S, spleen, L, liver, K, kidney, T, tumour. **c**, Average intensity of TIOs- Fe_3O_4 in the corresponding organs of the indicated tumour models at a series of time points. **d**, The tumour to liver ratio of MPI intensity at different time points. **e**, The MPI images of three independent CT26 tumour bearing mice intravenously injected with Fe_3O_4 -DHCA for 48 hours. Data are shown as mean \pm s.d. ($n = 3$, independent experiments). **f**, Tumor to liver ratio of TIOs in three different tumor bearing mice at different time points. **g**, Confocal images of catalytic amount of Dil labelled TIOs (red) treated Calcein-AM-

labelled BMDCs, iBMDMs, or TC-1, AML12, TCMK-1 (green). **h**, Western blot of the VCAM-1 expression in indicated cell lines. **i-l**, Inflammatory effects investigation of TIOs+NIR³ treated 4T1 tumour-bearing mice. **i**, TIOs+NIR³ treatment scheme in BALB/c mice implanted subcutaneously with 4T1 cells (3 mice per group). The tumour-bearing mice were intravenously injected with TIOs (iv, 5×10^6 units/mouse) and then treated with NIR light or not (980 nm, 0.5 W cm^{-2} , 5 min). **j**, The body weight of mice was measured at the indicated time. **k**, The rectal temperature of mice was measured at the indicated time. **l**, The levels of IL-6 in serum of mice were measured by ELISA at the indicated time.

Revision Data Figure 3. **a**, Characterize the ability of RSL3@azo-Ce6-mSiO₂-UCNPs in loading Ce6, azo, RSL3. Standard curve for concentration and absorbance of Ce6, azo and RSL3. Statistical table of the content of each component (Ce6, azo and RSL3) in RSL3@azo-Ce6-mSiO₂-UCNPs. **b-h**, Analysis of systemic inflammatory response in BALB/c female mice treated with TIOs. $n = 6$ mice for indicated groups. **b**, TIOs mediated Systemic inflammatory response scheme in BALB/c female mice. On day 1, 2, 3, 4 and 5 after intravenously injection of TIOs (5×10^6 units/mouse), blood was gained from the eyeballs of mice. **c-e**, The RBC, WBC and PLT of mice was measured at the indicated time. **f**, The rectal temperature of mice was measured at the indicated time. **g**, serum IL-6 concentration of mice. **h**, Body weight of mice. All measurements (n) are biologically independent. **i**, T-cell-inspired oncolytic system (TIOs) mediated cell lysis. The TIOs bound the tumour cells by antigens coordination. Then, the NIR light would trigger the TIOs to generate ROS and release RSL3 to destroy the tumour cells plasma membrane by lipids peroxidation. After the tumour cells lysis, the TIOs would regenerate and bind other tumour cells. Thus, the catalytic killing would happen when NIR light was inputted.

Revision Data Figure 4. **a**, TEM images of TIOs. Scale bars, 5 μm , 1 μm . **b**, The heating curve *in vitro*. The temperature of PBS, RSL3@azo-Ce6-mSiO₂-UCNPs and TIOs treated 4T1 tumour bearing mice with the indicated NIR light. Data shown as mean \pm s.d. ($n = 3$, independent experiments). **c**, The NIR light controlled ROS generation of TIOs. The TIOs were pre-treated with SOSG. The image of TIOs after

NIR light treatment. Scale bar, 20 μm . Flow cytometry analysis of ROS generation from TIOs by NIR light (980 nm, 1.0 W cm^{-2} , 5 min). The ROS was detected by SOSG probe with continues NIR light. **d**, The stability of TIOs in PBS at 25°C or 4°C. The quantity variation of TIOs was determined by cell counter assay (10^5 units / tube). Data are shown as mean \pm s.d. (n = 3, independent experiments). **e**, The whole-body thermal imaging of subcutaneous 4T1 tumour-bearing BALB/c female mice treated with TIOs (5×10^6 units/mouse), or PBS after NIR 980nm (0.5 Wcm^{-2} or 1.0 Wcm^{-2}) for 5 min or 10 min. Images were obtained using a thermal camera. n = 3 mice for indicated groups.

Revision Data Figure 5. a, Confocal images of the binding between DiI-labelled TIOs (red) and Calcein-AM-labelled healthy cells (green). Top, TC-1 cells. Middle, AML12 cells. Bottom, TCMK-1 cells. Scale bar, 100 μm . **b**, Cytotoxicity of RSL3@azo-Ce6- mSiO_2 -UCNPs to Raw 264.7 cells or BMDMs after incubation at 37 °C for indicated time points (6 h, 12 h, 24 h and 48 h). All data are shown as mean \pm s.e.m from three biological replicates. **c**, Cytotoxicity of NIR 980nm to tumour cells (4T1, CT26 and EMT6). Cell viability of NIR 980nm-treated tumour cells. All data are shown as mean \pm s.e.m from three biological replicates. **d**, Flow cytometry of PD-L1 in tumour cells (4T1, CT26 and EMT6).

Revision Data Figure 6. a, Cell morphology under the indicated conditions. For NIR light alone treatment, the tumour cells were irradiated with 1 Wcm^{-2} of NIR light for 5 min. For the Erastin +RSL3 treatment, before testing, Erastin (2 μM) and RSL3 (1 μM) were incubated with tumor cells for 24h. Scale bars, 40 μm . **b**, Representative immunoblots for GPX4 in the indicated tumour cells. **c**, Relative expression levels of GPX4/ β -actin in the indicated tumour cells. Data are shown as mean \pm s.e.m from three independent experiments; two-tailed unpaired Student's t-test was performed. **d**, Representative time-lapse images of TIOs-mediated EMT6 lytic cell death after NIR light (980 nm, 1.0 W cm^{-2}) for 5 min (top) and EMT6 treated with CD8⁺ T cells (bottom). White arrows show the lytic cell death morphology at representative time points. Scale bar, 100 μm .

Revision Data Figure 7. a, Representative fluorescence images of lipid peroxidation based on C11 581/591 BODIPY stained 4T1 tumours (Nu/Nu mice). TIOs+NIR treated

mice (980 nm, 0.5 W cm⁻², 5 min) (n = 3 Nu/Nu mice) or TIOs alone treated mice (n = 3 Nu/Nu mice). Scale bars, 10 μm. **b**, Mean fluorescence intensities (MFI) of oxidized BODIPY (FITC channel) or, non-oxidized BODIPY (PE channel) and the relative lipid ROS in 4T1 tumours (Nu/Nu mice) with indicated treatments. The relative lipid ROS are calculated as the ratio of oxidized and non-oxidized BODIPY MFI. Tumour lipid peroxidation of 4T1 tumor-bearing mice treated with PBS, TIOs+NIR (980 nm, 0.5 Wcm⁻², 5 min) or TIOs alone, n = 3 Nu/Nu mice for indicated groups. All data are shown as mean ± s.e.m from three biological replicates; two-tailed unpaired Student's t-test was performed. **c**, Flow cytometry analysis of C11 581/591 BODIPY fluorescence in 4T1 tumours (Nu/Nu mice) treated with TIOs+NIR or TIOs alone. **d**, The TIOs bound tumour cells were treated with or without NIR light (980 nm, 1.0 W cm⁻², 5 min). Red column, with NIR light. Black column, without NIR light. All data are shown as mean ± s.e.m from three biological replicates; two-tailed unpaired Student's t-test was performed.

Revision Data Figure 8. **a**, The VCAM-1 protein on the Fe₃O₄@DVB-VCAM-1 magnetic beads was detected by western blot. **b-c**, The ability of Fe₃O₄@DVB-VCAM-1 to bind Integrins was detected by western blot and flow cytometry. Integrins were added to the Fe₃O₄@DVB-VCAM-1 and Fe₃O₄@DVB-BSA, respectively. After incubation, centrifuging to collect sediment and washing. **d**, The Fe₃O₄@DVB-cy5-VCAM-1 bound Dil-labelled TIOs or Calcein-AM-labelled Raw 264.7 cells was detected by flow cytometry. **e**, Representative confocal images of the Fe₃O₄@DVB-cy5-VCAM-1 bound Calcein-AM-labelled Raw 264.7 cells. Scale bar, 20 μm. **f**, Representative confocal images of the Fe₃O₄@DVB-cy5-VCAM-1 bound Dil-labelled TIOs. Scale bar, 20 μm.

Materials and methods

Materials

Brilliant Violet 785™ anti-mouse CD274 (B7-H1, PD-L1) antibody was purchased from BioLegend. Purified NA/LE Hamster Anti-Mouse CD3e, Purified NA/LE Hamster Anti-Mouse CD28 and PE-CF594 hamster anti-mouse CD279 (PD-1) were purchased from

BD Biosciences. DiIC18(3) was obtained from Yeasen Biotechnology.

Cell lines and cell culture conditions

After preparing a single-cell suspension from mouse spleen, CD8⁺ T cells were negatively selected by IPHASE Mouse (ICR/CD1) CD8⁺ T Cells Isolation Kit (iPhase Biosciences). Isolated CD8⁺ T cells were cultured in 96-well flat bottom plates with precoated anti-CD3 (1 µg/mL; clone 145-2C11; BD Biosciences) and soluble anti-CD28 (1 µg/mL; clone 37.51; BD Biosciences) for 48 hours. CD8⁺ T cells were added with complete RPMI1640 media with 10% FBS, 1% penicillin, 100 µg/mL streptomycin, 1 mM sodium pyruvate (Gibco), and nonessential amino acids (Gibco).

Characterization of the TIOs

The morphology, size distribution and zeta potential of the nanoparticles or TIOs were characterized by scanning electron microscopy (SEM, FEI Teneo), transmission electron microscope (TEM, FEI Tecnai Spirit) and dynamic light scattering (DLS, Malvern ZetasizerNano S90). The amount of loaded Ce6 was determined by the difference between the initial amount of Ce6 and the amount of Ce6 in the supernatant after centrifugation. The absorbance of Ce6 was determined by UV-visible spectrophotometer, and quantified by the standard curve. The loading of azo and RSL3 in RSL3@azo-Ce6-mSiO₂-UCNP_s was also determined using the same method as above.

Stability of TIOs.

To examine the stability of TIOs, TIOs was resuspended in 1.0 mL PBS at 25°C or 4°C, dynamic light scattering (DLS, 724 Malvern Zetasizer Nano S90) combined with cell counter was used at specified times to assess the change of TIOs in the number.

Preparation and characterization of Fe₃O₄@DVB-VCAM-1

Preparation of Fe₃O₄@DVB. Fe₃O₄ superparamagnetic nanoparticles were synthesized by chemical coprecipitation, and some modifications were made. First,

12.5 mL FeCl_2 (0.12 mol/L) and 12.5 mL FeCl_3 (0.2 mol/L) were mixed and stirred for 30 min in a three-way bottle pre-vented with nitrogen. The mixture was then heated to 55 °C and NaOH (3 M) was quickly added to adjust the pH of the suspension to 10-12. The mixture was raised to 65°C and stirred for a further 40 min. Then slowly add 50 mL (0.4mol /L) sodium oleate solution to the mixture, heat up to 90 °C and stir for 30 min. After cooling to room temperature, it is cleaned with ultra-pure water under the action of an external magnet to remove excess sodium hydroxide and sodium oleate. The obtained Fe_3O_4 nanoparticles was dispersed and stored in an aqueous solution. 4 ml Fe_3O_4 nanoparticles (20 mg/mL), 3 g polyethylene glycol (PEG, $M_n=4000$), 30 mL ethanol and 20 mL ultra-pure water were mixed ultrasonic for 30 min. Add the monomer (8 mL styrene, 0.6 mL acrylic acid), the crosslinker (0.15 mL DVB) and the initiator ((benzoyl peroxide, BPO, 1.6 g) in sequence and stir for 40 min. The mixture was heated in an oil bath to 75°C for 10 h. At the end of the reaction, the solution was cooled to room temperature and washed three times with ethanol and ultra-pure water under the influence of an external magnet.

Preparation of $\text{Fe}_3\text{O}_4@DVB\text{-VCAM-1}$. VCAM-1 protein was conjugated to carboxylate microbeads by a two-step EDC/NHS coupling. 0.5mg $\text{Fe}_3\text{O}_4@DVB$ was washed 3 times with PBS under the action of an external magnet. Then 10 μL EDC (15 mg/mL, PBS) and 10 μL NHS (9 mg/mL, PBS) were added and incubated at room temperature for 35 min. After washing with 200 μL MES solution (0.05 mol/L, pH 5.0) for two times, 100 μL VCAM-1 protein solution (55 μg /mL) was added to activated microbead suspension, gently shaken at 37°C, and incubated again for 2.5 h. Magnetic beads are separated and supernatant and precipitates are collected under the action of an external magnetic field. After that, the magnetic beads were washed 3 times with PBS. Protein-coupled microbeads were re-suspended in 250 μL BSA solution (20% (w/v)) for 12 h in 4°C, blocking all remaining non-specific protein-binding sites. After separation with a magnet, the microbeads were stored in a refrigerator of 4°C until used. $\text{Fe}_3\text{O}_4@DVB\text{-BSA}$ prepared by a similar method described above. The $\text{Fe}_3\text{O}_4@DVB\text{-VCAM-1}$ and supernatants collected by the above methods were used to detect VCAM-1 protein connectivity by western blot. For western blot, separated

proteins were transferred to PVDF membranes (Merck Millipore). After 10 min blocking in protein-free rapid blocking solution (Epizyme Biotech), the membranes were incubated with Diluted anti-VCAM-1 (1:1000) overnight at 4°C. After washing 5 times with TBST, the membranes were further incubated with Anti-rabbit IgG, HRP-linked Antibody (1:5000) for 2 hours. After washing 5 times in TBST, the membranes were immersed in Thermo Scientific SuperSignal West Pico PLUS chemiluminescent substrate and exposed using an Amersham ImageQuant 800 (Cytiva).

Characterization of Fe₃O₄@DVB-VCAM-1. The binding effect of Fe₃O₄@DVB-VCAM-1 to Integrin α4 and Integrin β1 was detected by flow cytometry and western blot. 0.5 mg Fe₃O₄@DVB-VCAM-1 or Fe₃O₄@DVB-BSA and 5 µg Integrin α4 protein or 5 µg Integrin β1 protein were incubated on ice for 1 hour and then washed three times under an external magnetic field. The binding product were dispersed in 100 µL of cell staining buffer, and then 0.25 µg of the corresponding antibodies was added (anti-Integrin β1-PE and anti-Integrin α4-purified), incubating at 4°C for 20 min in the dark. Then, 100 µL Anti-rabbit IgG (H+L), F(ab)₂ Fragment (Alexa Fluor® 488 Conjugate) (CST, 1:1000) was added to purified antibodies, incubating at 4°C in the dark for 20 min. Western blot was used to detect the expression of Integrin α4 and Integrin β1, which was the same as the above method.

Fe₃O₄@DVB-VCAM-1 mimic the mechanism of tumour cell recognition by TIOs *in vitro*

Evaluation of the Fe₃O₄@DVB-VCAM-1 bound Raw 264.7 and TIOs. Raw 264.7 was seeded in glass-bottom dishes (10⁶ cells/dish) with Calcein-AM (10 µL, 1 mg ml⁻¹) containing cell culture medium for 6 h. The above Raw 264.7 was washed twice with PBS to generate the Calcein-AM labelled Raw 264.7 cells. The 0.5 mL Cy5-labeled Fe₃O₄@DVB-VCAM-1 (1 mg/mL) were incubated with the Calcein-AM labelled Raw 264.7 cells (10⁷ units/dish) and Dil labelled TIOs (10⁷ units/dish) for 1 h, respectively, and then washed with PBS for 3 times under the action of an external magnet. Representative images of Fe₃O₄@DVB-VCAM-1 bound Raw 264.7 cells and TIOs were observed and captured with *Nikon* confocal microscope(A1R-SIM-STORM). The

image data shown are representative of at least three randomly selected fields. At the same time, the precipitation and supernatant under the external magnet were collected by the same method as above, and the supernatant and the precipitate after washing with PBS for 3 times were analyzed by flow cytometry to analyze the combination of $\text{Fe}_3\text{O}_4@\text{DVB-VCAM-1}$ to Raw 264.7 and TIOs.

Preparation of TIOs- Fe_3O_4

Preparation of $\text{Fe}_3\text{O}_4\text{-DHCA}$. Monodisperse Fe_3O_4 was prepared from ferric acetylacetonate in oleic acid by high temperature thermal decomposition. Iron (III) acetylacetonate (12 mmol) and oleic acid (38 mmol) were added to benzyl ether (50 mL) and stirred magnetically under nitrogen for 30 minutes. The reaction mixture was slowly heated to 165°C for 30 min, followed by reflux at 280°C under nitrogen for another 30 min. Once the reaction is over, cool the black-brown mixture to room temperature. Clean the precipitate with ethanol 3 times and collect the precipitate under the condition of external magnet. The hydrophobic monodisperse Fe_3O_4 is transferred to the aqueous phase by ligand exchange reaction to obtain $\text{Fe}_3\text{O}_4\text{-DHCA}$. 50 mg DHCA was dissolved in 6 mL tetrahydrofuran (THF) in a three-necked flask (25 mL). The solution is heated to 50°C under argon. Then, 18 mg of monodisperse Fe_3O_4 was dispersed in 1 mL of THF and added drop by drop. After reaction for 3 h, it was cooled to room temperature, and $\text{Fe}_3\text{O}_4\text{-DHCA}$ were precipitated by adding 500 μL NaOH (0.5 M). Precipitation is collected by centrifugation (5000 rpm/min) and redispersed in water.

Preparation of TIOs- Fe_3O_4 . TIOs- Fe_3O_4 is prepared in a similar way to TIOs. The mixture of $\text{Fe}_3\text{O}_4\text{-DHCA}$ ($300 \mu\text{g ml}^{-1}$) and RSL3@azo-Ce6-mSiO₂-UCNPs ($300 \mu\text{g ml}^{-1}$) was incubated with Raw 264.7 cells ($\sim 10^7$ cells/dish) for 1, 2, 3, and 4 h and then gently washed in DMEM complete medium to produce TIOs- Fe_3O_4 precursors. The precursors of $\text{Fe}_3\text{O}_4\text{-TIOs}$ were soaked in liquid nitrogen for 24 h, defrosted at 37°C , centrifuged at 1000 rpm for 3 min, washed and then resuspended in DMEM complete medium or PBS to produce TIOs- Fe_3O_4 .

MPI imaging.

Tumour cells (4T1, EMT6, CT26, 1×10^6 cells) in 25 μ l PBS were implanted into the right flank of BALB/c female mice (6 weeks old). When the tumour volumes reached ~ 150 mm³ (7-10 days), the mice were randomly divided into five groups (n = 3 in every group), then TIOs-Fe₃O₄ (2×10^6 units/mouse) was injected intravenously. MPI imaging is performed using an MPI scanner (Magnetic Insight Inc, MOMENTUM). The frequency of MPI is 45 kilohertz. The magnetic gradient intensity of MPI is 0-5.5 T/m. During the experiment, mice were anesthetized with 2% isoflurane at 2 L/min oxygen flow. In addition, tumours and major organs (heart, liver, spleen, lungs, and kidneys) were collected at 6, 24, 48, 72, and 96 h after injection and imaged by an MPI scanner (Magnetic Insight Inc, MOMENTUM). The analysis of MPI imaging data processed by VivoQuant software 2.0.

***In vitro* tumour cell recognition**

Evaluation of TIOs recognizes BMDCs. BMDCs were seeded in glass-bottom dishes (1×10^6 cells/dish) with Calcein-AM (1 μ M) containing cell culture medium for 6 h. The cells were washed twice with PBS to make the Calcein-AM labelled BMDCs. The Dil labelled TIOs (5×10^6 units/dish) was added and co-incubating for 6 h, unbound TIOs were removed by washing with PBS. Representative images of TIOs bound tumour cells were observed and captured with a *Nikon* confocal microscope (A1R-SIM-STORM). The image data shown are representative of at least three randomly selected fields.

Systemic inflammatory response.

On day 1, 2, 3, 4 and 5 after intravenously injection of TIOs, blood was gained from the eyeballs of mice to detect RBC, WBC and PLT, and serum IL-6 concentration was detected by ELISA. In addition, the rectal temperature and body weight of mice were measured.

The release of HMGB1 and ATP.

To evaluate the immunogenic cell death induced by TIOs, the release of DAMP molecules (such as HMGB1 and ATP) were detected in vitro. The supernatant of TIOs bound tumour cells treated with NIR light (980 nm, 1.0 W cm⁻², 5 min) was collected and the release of HMGB1 was detected by ELISA. In addition, ATP release from TIOs bound tumour cells treated with NIR was tested using an ATP Assay Kit (Beyotime Biotechnology).

REVIEWERS' COMMENTS

Reviewer #1 (Remarks to the Author):

The authors have addressed my comments.

Reviewer #2 (Remarks to the Author):

The authors have addressed and revised the manuscript according to the comments. I recommended to accept this manuscript for publication.

Reviewer #3 (Remarks to the Author):

In this study, the authors, Zuo et al., embarked on the development and characterization of a micro-scale system inspired by cytotoxic T cells, coined as the TIO system. This novel platform is meticulously designed for cancer immunotherapy applications. Drawing on the intriguing proposition that chemically-modified macrophages, specifically RAW and BMDMs, can mimic the functionality of cytotoxic T cells, the authors presented compelling data to bolster this hypothesis. Taken as a whole, this study promises to become a stepping stone in the evolution of immunotherapeutic approaches to cancer treatment and this manuscript will be of interest to a broad audience in the cancer field.

The revised manuscript by Zuo et al. shows significant improvements. The authors have comprehensively addressed all of the reviewers' suggestions and incorporated new data to respond to key comments and concerns. The study is well-designed and effectively performed. I recommend accepting this work for publication. Minor typographical and grammatical edits can be addressed during the editorial stage.

Reviewer #4 (Remarks to the Author):

[none]

Point to point response:

Reviewer #1 (Remarks to the Author):

The authors have addressed my comments.

We sincerely appreciate all your efforts in reviewing the manuscripts and your suggestions have really improved the quality of our work!

Reviewer #2 (Remarks to the Author):

The authors have addressed and revised the manuscript according to the comments. I recommended to accept this manuscript for publication.

We sincerely appreciate all your efforts in reviewing the manuscripts and your suggestions have really improved the quality of our work!

Reviewer #3 (Remarks to the Author):

In this study, the authors, Zuo et al., embarked on the development and characterization of a micro-scale system inspired by cytotoxic T cells, coined as the TIO system. This novel platform is meticulously designed for cancer immunotherapy applications. Drawing on the intriguing proposition that chemically-modified macrophages, specifically RAW and BMDMs, can mimic the functionality of cytotoxic T cells, the authors presented compelling data to bolster this hypothesis. Taken as a whole, this study promises to become a stepping stone in the evolution of immunotherapeutic approaches to cancer treatment and this manuscript will be of interest to a broad audience in the cancer field.

The revised manuscript by Zuo et al. shows significant improvements. The authors have comprehensively addressed all of the reviewers' suggestions and incorporated new data to respond to key comments and concerns. The study is well-designed and effectively performed. I recommend accepting this work for publication. Minor typographical and grammatical edits can be addressed during the editorial stage.

We sincerely appreciate all your efforts in reviewing the manuscripts and your suggestions have really improved the quality of our work!

Reviewer #4 (Remarks to the Author):

[none]

We sincerely appreciate all your efforts in reviewing the manuscripts and your suggestions have really improved the quality of our work!